# MADCluster: Model-agnostic Anomaly Detection with Self-supervised Clustering Network

## Abstract

In this paper, we propose MADCluster, a novel model-agnostic anomaly detection framework utilizing self-supervised clustering. MADCluster is applicable to various deep learning architectures and addresses the 'hypersphere collapse' problem inherent in existing deep learning-based anomaly detection methods. The core idea is to cluster normal pattern data into a 'single cluster' while simultaneously learning the cluster center and mapping data close to this center. Also, to improve expressiveness and enable effective single clustering, we propose a new 'One-directed Adaptive loss'. The optimization of this loss is mathematically proven. MADCluster consists of three main components: Base Embedder capturing high-dimensional temporal dynamics, Cluster Distance Mapping, and Sequence-wise Clustering for continuous center updates. Its model-agnostic characteristics are achieved by applying various architectures to the Base Embedder. Experiments on four time series benchmark datasets demonstrate that applying MADCluster improves the overall performance of comparative models. In conclusion, the compatibility of MADCluster shows potential for enhancing model performance across various architectures.

## 1 Introduction

In modern infrastructures such as industrial equipment and data centers, numerous sensors operate continuously, generating and collecting substantial amounts of continuous measurement data. Effective detection of abnormal system patterns through real-time monitoring in these large-scale systems helps prevent significant monetary losses and potential threats (Djurdjanovic et al., 2003; Leon et al., 2007; Yang et al., 2021b). However, detecting anomalies in complex time-series systems is challenging due to factors such as the diversity of abnormal patterns (irregular, unusual, inconsistent, or missing data) (Ruff et al., 2021), temporal dependencies of adjacent data, and the complexity where boundaries between normal and abnormal can be ambiguous (Yang et al., 2021b). Moreover, anomalies are generally rare, making it difficult to obtain labels and thus challenging to apply supervised or semi-supervised learning methods (Yang et al., 2021a). Researchers have designed various time-series anomaly detection methods to address these issues. In unlabeled environments, unsupervised learning is primarily used over supervised and semi-supervised learning. Classical unsupervised learning-based methods include density estimation methods (Parzen, 1962; Bishop, 1994; Breunig et al., 2000), kernel-based methods (Schölkopf et al., 2001; Tax & Duin, 2004), while deep learning-based unsupervised methods include clustering-based (Zong et al., 2018) and deep one-class classification-based approaches (Ruff et al., 2018; Hojjati & Armanfard, 2023; Shen et al., 2020).

Deep one-class classification-based methods learn normal patterns of complex high-dimensional data and identify the boundaries of normal data in feature space. The main goal of these methods is to find a minimum volume region (e.g., hypersphere or hyperplane) that contains normal data, thereby detecting anomalies as data points that fall outside the learned boundary. These unsupervised anomaly detection algorithms are gaining attention due to their powerful representation learning capabilities for complex high-dimensional data and their ability to effectively model the distribution of normal data. Moreover, from the perspective of improving performance through integration with other models, one-class classification methods can be seen as model-agnostic methodologies appli-

cable to various models. For example, in Log Anomaly detection tasks, they are used as an objective function to map embeddings of normal data near the normal center ((Guo et al., 2021), (Almodovar et al., 2024)). However, these methods may face the 'hypersphere collapse' problem, a persistent issue in one-class classification where network weights converge to a trivial solution of all zeros. This leads to the problem of falling into local optima rather than global optima due to the limited expressiveness of weights in the feature space.

In this paper, we propose the **M**odel-agnostic **A**nomaly **D**etection with self-supervised **Cluster**ing network called **MADCluster**, which is applicable to existing deep learning anomaly detection models and solves the hypersphere collapse problem. The core idea of MADCluster is to cluster normal pattern data into a single cluster while simultaneously learning the cluster center and mapping data close to this center. This is motivated by the desire to achieve model-agnostic characteristics without constraints on expressiveness in the feature space. Specifically, we propose a structure with two modules: a distance mapping module and a clustering module. The first is a distance mapping module for mapping normal data near the center, and the second is a clustering module that learns central coordinates by single-clustering normal data. In particular, for the clustering module, we newly define an 'One-directed Adaptive loss' for effective single clustering and provide a proof of optimization for this One-directed Adaptive loss. The main contributions of MADCluster are summarized as follows:

- *Model-Agnostic Methodology*: MADCluster model-agnostic nature ensures compatibility with a wide range of deep neural network-based models, thus overcoming the limitations of specific network architectures. MADCluster offers improved performance and adaptability across diverse analytical scenarios. Unlike model-specific anomaly detection methods, MADCluster proposes a more flexible and universally applicable approach.

- *Preventing Hypersphere Collapse*: MADCluster, a clustering-based anomaly detection method, effectively addresses the hypersphere collapse problem. It distinctively updates central coordinates through network parameters, efficiently preventing the all-zero parameter problem and enabling richer representational power in the feature space.

- *Optimization Proof for Single Clustering*: MADCluster enables more accurate clustering when performing single clustering for anomaly detection tasks by simultaneously learning the cluster center and decision boundary. We provide a mathematical proof for optimizing the One-directed Adaptive loss ensures the theoretical soundness of this method, providing a robust foundation for its practical application.

- *Performance on Public Datasets*: Despite its simple structure, the anomaly detection model applying MADCluster demonstrates improved performance on four real-world benchmark datasets compared to existing methods. It is noteworthy that the model simplicity does not compromise its effectiveness. Furthermore, there is considerable potential for performance enhancement if various techniques that are more complex and more effective at feature extraction are integrated.

## 2    RELATED WORK

*Anomaly Detection.* Classical anomaly detection methods have explored the unsupervised learning paradigm, including density estimation methods such as Local Outlier Factor (LOF) (Breunig et al., 2000), kernel-based methods like One-Class SVM (OC-SVM) (Schölkopf et al., 2001), and Support Vector Data Description (SVDD) (Tax & Duin, 2004). These methods typically assume that the majority of the training data represents normal conditions, enabling the model to capture and learn these characteristics. Anomalies are detected when new observations do not conform well to the established model (Chen et al., 2001; Liu et al., 2013; Zhao et al., 2013). Recent advances in deep learning (LeCun et al., 2015; Schmidhuber, 2015) have led to attempts to integrate the powerful representation learning capabilities of deep networks into traditional classifiers. For example, DAGMM (Zong et al., 2018) combines Gaussian Mixture Model (GMM) with Deep Autoencoder, and DeepSVDD (Ruff et al., 2018) replaces the kernel-based feature space with a feature space learned by deep networks. However, DeepSVDD faces a significant issue known as hypersphere collapse, where the network weights converge to a trivial solution of all zeros (Ruff et al., 2018). To mitigate this, modifications such as fixing the hypersphere center and setting the bias to zero have been implemented. While these measures help prevent hypersphere collapse, they can limit

the overall performance and effectiveness of the algorithm. In recent years, several studies have proposed solutions to the hypersphere collapse problem. DASVDD (Hojjati & Armanfard, 2023) is structured as an autoencoder network. It involves fixing the hypersphere center $c$ to train the encoder and decoder, and then fixing the network parameters to learn the hypersphere center $c$ based on latent representations. This approach jointly trains the autoencoder and SVDD to update $c$. The Temporal Hierarchical One-class (THOC) model (Shen et al., 2020) updates the center coordinates by mapping multi-scale temporal embeddings at various resolutions near multiple hyperspheres, clustering features from all intermediate layers of the network. Both methods address the hypersphere collapse by updating the center $c$.

*Clustering.* Clustering is a data mining technique that aids in discovering and understanding natural structures in large datasets. The primary goal of clustering is to group data points with similar characteristics, thereby identifying inherent patterns and structures within the data (Pavithra & Parvathi, 2017). Traditional clustering methods include density-based clustering (Ester et al., 1996; Comaniciu & Meer, 2002) and distribution-based clustering (Bishop, 2006). These methods are effective when features are relevant and representative in finding clusters. However, they struggle to cluster high-dimensional complex data effectively as the dimensionality increases, leading to a decrease in the significance of distance measurements (Pavithra & Parvathi, 2017; Ren et al., 2024). To map complex data into a feature space conducive to clustering, many clustering methods focus on feature extraction or feature transformation, such as PCA (Wold et al., 1987), kernel methods (Hearst et al., 1998), and deep neural networks (Liu et al., 2017). Among these methods, deep neural networks represent a promising approach due to their excellent nonlinear mapping capabilities and flexibility. Deep Embedded Clustering (DEC) (Xie et al., 2016) is a methodology that utilizes an autoencoder structure to learn low-dimensional representations of data and perform clustering based on these representations. Specifically, DEC defines a clustering objective function using soft cluster assignments and an auxiliary target distribution, optimizing network parameters and cluster centers while minimizing this function. However, DEC optimizes using only the clustering loss function, making it difficult to maintain important local structures of the data, potentially distorting the learned feature space. Improved Deep Embedded Clustering (IDEC) (Guo et al., 2017) simultaneously optimizes clustering loss and reconstruction loss, enabling it to learn features while preserving the local structure of the data. Proposed method allows for consideration of both the overall cluster structure and local data relationships.

## 3 METHOD

In monitoring a system, we sequentially record $d$ measurements at regular intervals. In the context of time-series anomaly detection, we are given a set of time-series $\mathcal{X} = \{x_1, x_2, \ldots, x_T\}$, where each point $x_t \in \mathbb{R}^d$ indicates the observation at time $t$. The goal is to detect anomalies in periodic observations to identify any deviations from normal behavior. Detecting anomalies in time-series systems presents challenges such as temporal dependencies and pattern diversity, which is why we focus on time-series anomaly detection in an unsupervised learning setting.

We have developed the model-agnostic anomaly detection with self-supervised clustering (MADCluster) network for unsupervised time-series anomaly detection, addressing the aforementioned hypersphere collapse problem while maintaining model-agnostic characteristics. MADCluster leverages the self-learning technique to update the center of the normal cluster, mapping data closer to the updated centroid and minimizing the hypersphere in the feature space. Proposed method, using dynamic centers instead of fixed ones, enables more diverse and richer representations in the feature space, thereby enhancing anomaly detection performance. Therefore, due to its model-agnostic design, MADCluster can be applied to various deep learning architectures to improve performance, and as a lightweight model with fewer parameters and faster computational speed, it poses minimal burden in terms of time cost.

### 3.1 OVERALL ARCHITECTURE

Figure 1 illustrates the overall architecture of MADCluster, which consists of three main components: Base Embedder module, Sequence-wise Cluster module, and Cluster Distance Mapping module. On the left side, Base Embedder (section 3.1.1) initially processes the input to extract high-dimensional temporal dynamics. Extracted features are then fed into two modules on the right:

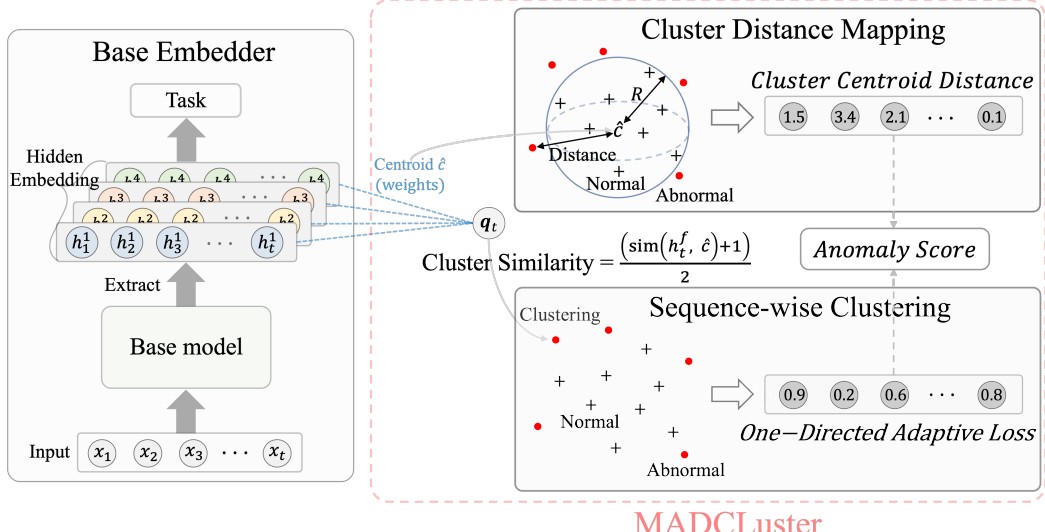

Figure 1: The proposed Model-agnostic Anomaly Detection with self-supervised Clustering (MAD-Cluster) network architecture. Base Embedder captures high-dimensional temporal dynamics. Output of Base Embedder, denoted as $h_t^f$, is fed into Cluster Distance Mapping and Sequence-wise Clustering modules.

Cluster Distance Mapping (section 3.1.2) and Sequence-wise Cluster (section 3.1.3). Cluster Distance Mapping module projects data from data space into feature space, concentrating it near the center coordinates. Sequence-wise Cluster module calculates cluster similarity for each instance and computes a One-directed Adaptive loss to update the center coordinates. Outputs of these two modules are combined through element-wise summation, which can be utilized either as an anomaly score itself or added to the anomaly score of the base model.

### 3.1.1 BASE EMBEDDER

To effectively detect anomalies in time-series data, it is crucial to extract the temporal characteristics of the data well. In the Base Embedder, we use the Dilated Recurrent Neural Network (D-RNN) (Chang et al., 2017) as the base model, which is designed to efficiently extract multi-scale temporal features from the time series data. D-RNN employs skip connections and dilated convolutions, allowing it to capture long-term dependencies and diverse temporal patterns across different time scales. The base model is not limited; it can utilize other anomaly detection models as well, all of which aim to extract complex hidden temporal dynamics within the data. When we consider a scenario where each process handles an input time series of length $T$, denoted as $\mathcal{X} \in \mathbb{R}^{d \times T}$, the extracted dynamics are formalized as follows:

$$h_t^f = \mathcal{F}_{\text{base\_model}}(x_t),$$ (1)

The output of the base model at time $t$, denoted as $h_t^f \in \mathbb{R}^{f \times 1}$, where $f$ represents dimensionality of the hidden feature space, reflects the learned features and extracted temporal dynamics. This flexible approach allows for the use of various models that can effectively capture the underlying temporal patterns in the data.

### 3.1.2 CLUSTER DISTANCE MAPPING

The MADCluster measures the deviation of the high-dimensional temporal dynamics $h_t^f$ from the cluster center $\hat{c}$. Unlike DeepSVDD, where the center is a pre-determined fixed point, MADCluster considers $\hat{c}$ as a learnable parameter. The objective for Cluster Distance Mapping is expressed as follows:

$$\mathcal{L}_{\text{distance}} = R^2 + \frac{1}{\rho} \sum_{t=1}^{T} \max\left\{0, \|\text{NN}(x_t; \mathcal{W}) - \hat{c}\|^2 - R^2\right\} + \lambda \Omega(\mathcal{W}). \tag{2}$$

In this case, $\text{NN}(x_t; \mathcal{W}) = h_t^f$, where $\text{NN}(\cdot; \mathcal{W})$ represents a Base Embedder with parameters $\mathcal{W}$. $\Omega(\mathcal{W})$ is a regularizer (such as the $l_2$-regularizer) and $\rho \in (0, 1]$ is a hyperparameter that balances the penalties against the sphere volume. $R$ is the radius and $\lambda$ is the learning rate. $R$ is determined based on the neural network output and the given hyperparameter $\nu$, rather than being a parameter. Instead, $R$ is computed using a specific quantile of the neural network outputs and the data loss values.

The goal is to minimize the distance loss function $\mathcal{L}_{\text{distance}}$ with respect to the neural network weights $\mathcal{W}$ and the cluster center parameters $\hat{c}$. If $\mathcal{L}_{\text{distance}}$ is updated without updating the center coordinates $\hat{c}$ through Sequence-wise Clustering, it may lead to hypersphere collapse. To mitigate this issue, MADCluster utilizes Sequence-wise Clustering to update $\hat{c}$, ensuring a continuously evolving centroid that accurately reflects the 'normal' data distribution. The cluster center can be viewed as the parameters that the Sequence-wise Clustering network needs to learn. The learning process is designed to ensure that each temporal feature embedding is closely mapped to the cluster center.

### 3.1.3 SEQUENCE-WISE CLUSTERING

In our Sequence-wise Clustering approach for anomaly detection in time-series data, we primarily focus on a single cluster representing 'normal' data. Data points are classified as normal if they exhibit a high similarity of belonging to this cluster, and abnormal otherwise. While our method shares similarities with DEC (Xie et al., 2016) in its use of self-learning for soft assignment, it diverges significantly in its approach to single clustering. Unlike conventional DEC, we discard the student's $t$-distribution, instead employing cosine similarity and a one-directed threshold to generate labels for single clustering. When the number of clusters is $k$, the clusters are denoted as $\{\hat{c}_j \in \mathbb{R}^f\}_{j=1}^{k}$. For scenarios with a single cluster center ($k = 1$), we avoid using the student's $t$-distribution. In a single-cluster scenario typical of anomaly detection tasks, the student's $t$-distribution would yield a constant similarity value of 1, resulting in ineffective learning of the cluster centroid. By modifying the similarity function for soft assignment, our Sequence-wise Clustering method enables a more focused approach on the single cluster representing normal data.

Sequence-wise Clustering conducts soft assignment and auxiliary target assignment. Soft assignment calculates a cluster auxiliary distribution for each temporal feature embedding. Then, auxiliary target assignment assigns cluster labels based on a learnable one-directed threshold parameter. Sequence-wise Clustering actively performs the learning process by comparing target labels with the auxiliary distribution, in order to train closely with the normal cluster.

**Step 1 (Soft Assignment):** We used cosine similarity as the metric to compare high-dimensional temporal dynamics $h_t^f$ from Base Embedder with the centroid vector $\hat{c} \in \mathbb{R}^{f \times 1}$, where $\hat{c}$ is a learnable parameter. This decision enables effective centroid learning and enables our model to differentiate between normal and abnormal data in a simplified single cluster approach. The cosine similarity between high-dimensional temporal dynamics $h_t^f$ at time $t$ and the centroid vector $\hat{c}$ is computed as:

$$q_t = \frac{(h_t^f)^{\top} \cdot \hat{c}}{\|h_t^f\|\|\hat{c}\|}, \tag{3}$$

$q \in \mathbb{R}^{T \times 1}$ indicates the soft assignment similarity, and $q_t$ is subsequently normalized to a range of $0 \le q_t \le 1$, through the transformation $q_t = \frac{q_t + 1}{2}$.

**Step 2 (Auxiliary Target Assignment):** The soft assignment similarity $q_t$ is normalized and then classified into binary categories based on a one-directed threshold $\nu$ to obtain the auxiliary target. The auxiliary target is calculated as follows:

$$p_t = \begin{cases} 1 & \text{if } q_t \ge \nu, \\ 0 & \text{otherwise,} \end{cases} \quad \text{s.t.} \quad 0 < \nu < 1 \tag{4}$$

$p \in \mathbb{R}^{T \times 1}$ plays the role of actual labels, and cluster center $\hat{c}$ and one-directed threshold $\nu$ are trained according to the difference between the similarity of belonging to the normal cluster, represented by $q_t$, and the auxiliary distribution $p_t$.

**One-directed Adaptive loss function:** We introduce a novel loss function called the One-directed Adaptive loss function. Through this proposed loss function, the one-directed threshold $\nu$ is trained to increase in value as learning progresses. The One-directed Adaptive loss function is defined as:

$$\mathcal{L}_{\text{cluster}} = -\sum_{t=1}^{T} p_t \log \left[ \frac{1 - \nu^{1-\nu}}{1 - \nu}(q_t - 1) + 1 \right] + (1 - p_t) \log \left[ q_t^{1-\nu} \right]. \tag{5}$$

The One-directed Adaptive loss function has the following characteristics: First, when the value of $q_t$ is fixed, the value of $\nu$ must increase to reduce the total loss, meaning the threshold increases as it is learned. Second, the distribution of $q_t$ should approach 1, not 0, during the learning process. Calculating the derivatives $\frac{\partial \mathcal{L}_{\text{cluster}}}{\partial q_t}$ and $\frac{\partial \mathcal{L}_{\text{cluster}}}{\partial \nu}$ shows that the loss $\mathcal{L}_{\text{cluster}}$ decreases as $q_t$ and $\nu$ increase, and a detailed explanation of this is provided in appendix A.

**Objective Function**: In MADCluster, the total objective function is a sum of the losses from Cluster Distance Mapping and Sequence-wise Clustering, and it is defined as follows:

$$\mathcal{L}_{\text{total}} = \mathcal{L}_{\text{distance}} + \mathcal{L}_{\text{cluster}}. \tag{6}$$

The entire procedure is detailed in Algorithm 1.

---

**Algorithm 1** **M**odel-agnostic **A**nomaly **D**etection with self-supervised **Cluster**ing network

---

**Require:** time-series $\mathcal{X} = \{x_1, x_2, \ldots, x_T\}$

1: **repeat**
2:     **for** each time step $t$ in $\mathcal{X}$ **do**
3:         Process $x_t$ using Base Embedder to get $h_t^f$
4:         Compute cosine similarity $q_t$ between $h_t^f$ and $\hat{c}$
5:         Normalize $q_t$ to range [0, 1]
6:         Assign auxiliary target $p_t$ by thresholding $q_t$ with $\nu$
7:     **end for**
8:     Compute $\mathcal{L}_{\text{distance}}$
9:     Compute $\mathcal{L}_{\text{cluster}}$
10:    Set $\mathcal{L}_{\text{total}} = \mathcal{L}_{\text{distance}} + \mathcal{L}_{\text{cluster}}$
11:    Update $\mathcal{W}$, $\hat{c}$, and $\nu$ based on $\mathcal{L}_{\text{total}}$ using backpropagation
12: **until** convergence

---

**Anomaly Score**: For a given time-series $\mathcal{X}$, consider an unseen observation at time $t$, denoted as $x_t$. The anomaly score is defined as:

$$\begin{aligned}
\text{Anomaly Score}(x_t) = -&\left\{ p_t \log \left[ \frac{1 - \nu^{1-\nu}}{1 - \nu}(q_t - 1) + 1 \right] + (1 - p_t) \log \left[ q_t^{1-\nu} \right] \right\} \\
&+ \left\| h_t^f - c^* \right\|^2 - R^2.
\end{aligned} \tag{7}$$

In this case, $c^*$ represents the cluster center of the trained model, and $\text{Anomaly Score}(x_t) \in \mathbb{R}^{T \times 1}$ serves as the point-wise anomaly score for $\mathcal{X}$. The anomaly threshold is determined using the percentile method based on the distribution of anomaly scores. Specifically, we set the threshold as the $(100 - \alpha)$-th percentile of the anomaly scores, where $\alpha$ is the expected anomaly ratio. An observation $x_t$ is labeled as abnormal if $\text{Anomaly Score}(x_t)$ exceeds anomaly threshold, and normal otherwise.

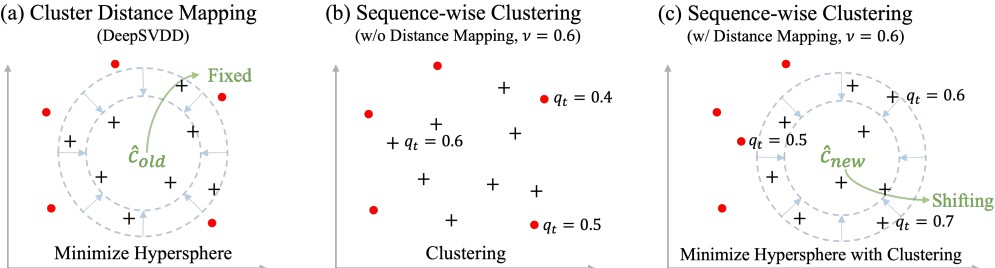

Figure 2: Comparison of anomaly detection approaches: (a) Cluster Distance Mapping, (b) Sequence-wise Clustering without Distance Mapping, and (c) Proposed approach combining Cluster Distance Mapping and Sequence-wise Clustering.

Finally, to provide an intuitive understanding of the mechanism behind our proposed method, Figure 2 illustrates the key differences between our approach and existing techniques. This visual comparison demonstrates how our method integrates the strengths of both Cluster Distance Mapping and Sequence-wise Clustering, addressing the limitations of each approach. Red dots represent potential anomalies, black plus-sign are normal data points, and the blue circle indicates the learned hypersphere.

1. **Cluster Distance Mapping (DeepSVDD):** This approach uses a fixed center coordinate $\hat{c}_{old}$ and minimizes the hypersphere radius $R$ to map data points close to the center. While the hypersphere shrinks around the fixed center, it potentially constrains data to cluster around a suboptimal point in the feature space.

2. **Sequence-wise Clustering (without Distance Mapping):** This method computes the similarity $q_t$ between the Base Embedder output $h_t^f$ and the center coordinate $\hat{c}$, then performs labeling based on a threshold $\nu$. Data points with similarity $q_t$ below the threshold are classified as anomalies. As shown, anomalies are scattered sporadically, indicating that this approach fails to capture local information effectively, potentially leading to inconsistent labeling of similar data points.

3. **Combined Cluster Distance Mapping and Sequence-wise Clustering:** By integrating both techniques, our method achieves several advantages. The center coordinate $\hat{c}_{old}$ is learned and shifts to a position $\hat{c}_{new}$ with richer representational power. The hypersphere is then minimized around this new center. Simultaneously, the approach incorporates local information, ensuring that similar data points are consistently labeled as normal or abnormal. Unlike the scattered anomalies in (b), our approach in (c) reflects local information, resulting in anomaly predictions that are more coherent within similar regions of the data space.

## 4 EXPERIMENTS

### 4.1 DATASETS

Description of the five experiment datasets: (1) PSM (Pooled Server Metrics, (Abdulaal et al., 2021)) is collected internally from multiple application server nodes at eBay with 26 dimensions. (2) Both MSL (Mars Science Laboratory rover) and SMAP (Soil Moisture Active Passive satellite) are public datasets from NASA (Hundman et al., 2018) with 55 and 25 dimensions respectively, which contain the telemetry anomaly data derived from the Incident Surprise Anomaly (ISA) reports of spacecraft monitoring systems. (3) SWaT (Secure Water Treatment, (Mathur & Tippenhauer, 2016)) data, which is collected from a water treatment testbed over 11 days. It is obtained from 51 sensors of the critical infrastructure system under continuous operations. During the training process, 20% of the training data was used for evaluation. The statistical details of the five benchmark datasets are summarized in Table 5 in appendix D.

## 4.2 IMPLEMENTATION DETAILS

Following the established protocols as outlined in previous studies (Shen et al., 2020; Xu et al., 2021), with a fixed window size of 100 for all datasets. Anomalies are identified among time points when their anomaly score, as defined in Equation equation 7, exceeds a specific threshold denoted as $\delta$. Our approach aligns with a widely-adopted adjustment strategy (Xu et al., 2018; Su et al., 2019; Shen et al., 2020; Xu et al., 2021): if a time point within a consecutive abnormal segment is marked as anomalous, we consider all anomalies within that segment as correctly detected. This strategy is based on the real-world observation that an anomalous time point often triggers an alert, directing attention to the entire segment. During the experiments conducted for MADCluster, we addressed over-confidence in the output $p_t$ resulting from Sequence-wise Clustering by applying label-smoothing. The smoothing process modifies the original label $p_t$ by applying a factor $\tau$ which serves to soften the label. The softened label $p_t$ is computed using the formula $p_t = p_t \times (1-\tau) + (1-p_t) \times \tau$. In this context, $\tau$ is the smoothing factor that is constrained by the condition $0 \leq \tau \leq 0.5$, facilitating the transition of $p_t$ from a hard to a soft label. We extensively compare our model with 11 baselines, including the reconstruction based models: USAD (Audibert et al., 2020), Anomaly Transformer (Xu et al., 2021), DCdetector (Yang et al., 2023); the density estimation models: LOF (Breunig et al., 2000) ; the clustering based methods: DeepSVDD (Ruff et al., 2018), ITAD (Shin et al., 2020), THOC (Shen et al., 2020); the autoregression based models: VAR (Anderson, 1976); the classic methods: OC-SVM (Tax & Duin, 2004), IsolationForest (Tony Liu et al., 2008); the sequential data processing models: D-RNN (Chang et al., 2017).

## 4.3 QUANTITATIVE RESULTS

Table 1 shows the evaluation results before and after applying MADCluster to 11 baseline models across four real-world datasets: MSL, SMAP, SWaT, and PSM. The proposed model improved the balance between precision and recall. Notably, the D-RNN model on MSL saw a 13.6% F1 score increase (81.24 to 94.84), due to a 23.26% improvement in recall. Similarly, the USAD model on PSM showed a 16.65% F1 score increase, also from improved recall. Conversely, the THOC model on SWaT and PSM had slightly decreased recall but substantially increased precision, improving overall performance. Except for these cases, all models showed increased recall. In summary, all models demonstrated enhanced F1 scores after applying MADCluster, with lower-performing models showing more significant improvements in recall.

Table 1: Performance metrics (Precision, Recall, F1-Score) for 11 models before and after applying MADCluster on four datasets. Results are in percentages, with best results in bold.

| DATASET | MSL | | | SMAP | | | SWaT | | | PSM | | |
|---|---|---|---|---|---|---|---|---|---|---|---|---|
| METRIC | P | R | F1 | P | R | F1 | P | R | F1 | P | R | F1 |
| OC-SVM | 59.78 | 86.97 | 70.82 | 53.85 | 59.07 | 56.34 | 45.39 | 49.22 | 47.23 | 62.75 | 80.89 | 70.67 |
| IF | 53.94 | 86.54 | 66.45 | 52.39 | 59.07 | 55.53 | 49.29 | 44.95 | 47.02 | 76.09 | 92.45 | 83.48 |
| LOF | 47.72 | 85.25 | 61.18 | 58.93 | 56.33 | 57.60 | 72.15 | 65.43 | 68.62 | 57.89 | 90.49 | 70.61 |
| VAR | 74.68 | 81.42 | 77.90 | 81.38 | 53.88 | 64.83 | 81.59 | 60.29 | 69.34 | 90.71 | 83.82 | 87.13 |
| ITAD | 69.44 | 84.09 | 76.07 | 82.42 | 66.89 | 73.85 | 63.13 | 52.08 | 57.08 | 72.80 | 64.02 | 68.13 |
| D-RNN | 88.88 | 74.81 | 81.24 | 93.58 | 99.29 | 96.35 | 78.59 | 100.00 | 88.01 | 97.59 | 96.52 | 97.05 |
| + MADCLUSTER | 91.83 | 98.07 | **94.84** | 93.58 | 99.36 | **96.39** | 93.02 | 100.00 | **96.39** | 97.42 | 97.94 | **97.68** |
| USAD | 92.47 | 86.03 | 89.13 | 93.51 | 94.26 | 93.88 | 94.41 | 75.93 | 84.16 | 97.61 | 68.66 | 80.62 |
| + MADCLUSTER | 92.99 | 94.46 | **93.72** | 93.64 | 99.24 | **96.36** | 99.44 | 77.06 | **86.83** | 97.61 | 96.94 | **97.27** |
| THOC | 88.45 | 90.97 | 89.69 | 92.06 | 89.34 | 90.68 | 83.94 | 86.36 | 85.13 | 88.14 | 90.99 | 89.54 |
| + MADCLUSTER | 91.87 | 95.74 | **93.76** | 93.07 | 92.36 | **92.71** | 92.63 | 83.80 | **87.99** | 96.82 | 88.05 | **92.23** |
| ANOTRANS | 91.92 | 96.03 | 93.93 | 93.59 | 99.41 | 96.41 | 89.10 | 99.28 | 94.22 | 96.94 | 97.81 | 97.37 |
| + MADCLUSTER | 92.05 | 97.93 | **94.90** | 93.64 | 99.50 | **96.48** | 93.25 | 100.00 | **96.51** | 97.42 | 98.59 | **98.00** |
| DCDETECTOR | 92.37 | 97.34 | 94.79 | 94.94 | 97.81 | 96.35 | 93.08 | 100.00 | 96.41 | 97.19 | 98.34 | 97.76 |
| + MADCLUSTER | 92.60 | 97.90 | **95.18** | 94.39 | 99.04 | **96.66** | 93.18 | 100.00 | **96.47** | 97.23 | 98.99 | **98.10** |

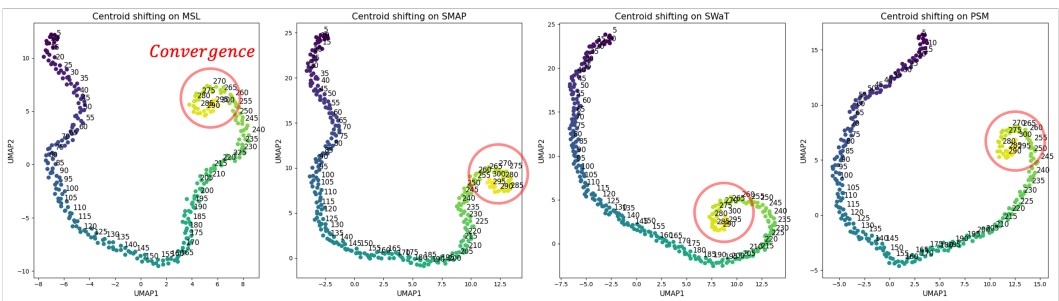

Figure 3: Visualization of centroid movement, captured every 5 epochs using UMAP.

## 4.4 QUALITATIVE RESULTS

We have addressed the limitations of previous models, particularly the issue of fixed center coordinates, through our proposed method, MADCluster. To visualize how the center coordinates move and converge, we employed UMAP (McInnes et al., 2018), a dimensional reduction technique, to represent the high-dimensional centroid in two-dimensional space. Figure 3 presents the two-dimensional mapping results across four datasets. This figure illustrates the evolution of cluster center coordinates, updated through MADCluster, visualized in two dimensions over 300 epochs. Throughout the training process, we observe that the cluster center converges towards specific points, exhibiting vibrating behavior within the converged area. This convergence, as opposed to divergence, indicates that the center coordinates are learning to represent more complex feature spaces. In Figure 4 to verify the effectiveness of the moving center coordinates during training and provide an intuitive understanding, we conducted a visual comparison between DeepSVDD and MADCluster.

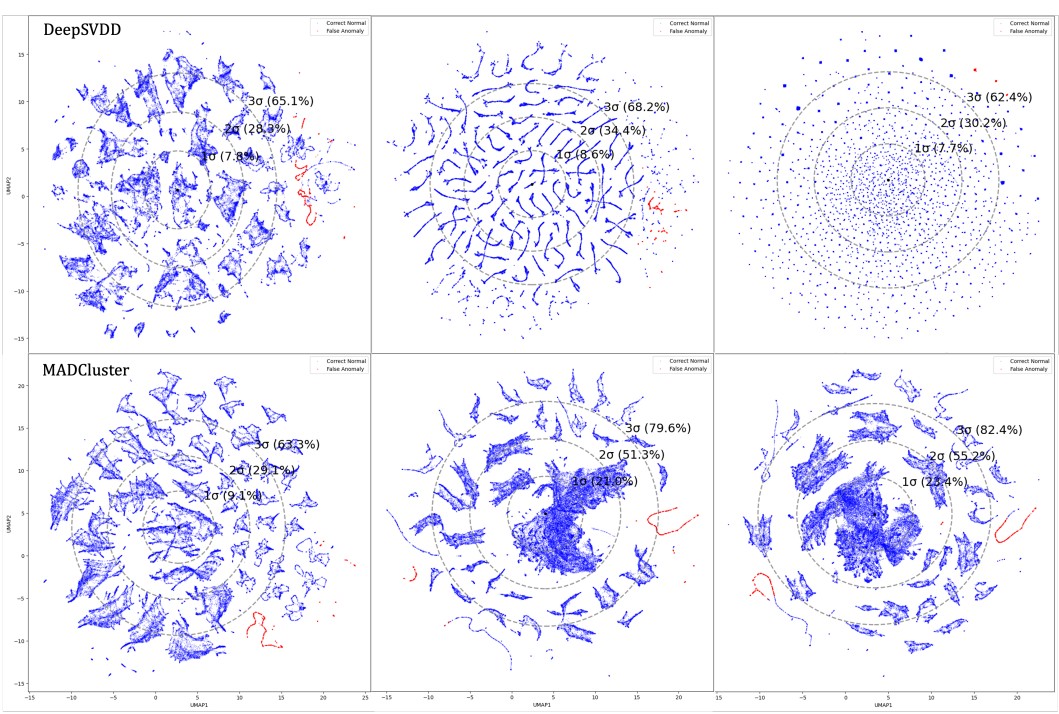

Figure 4: Hidden embedding visualization for DeepSVDD (top) and MADCluster (bottom) at epochs 1, 150, and 300. $\sigma$ represents the standard deviation from the center of hidden embeddings.

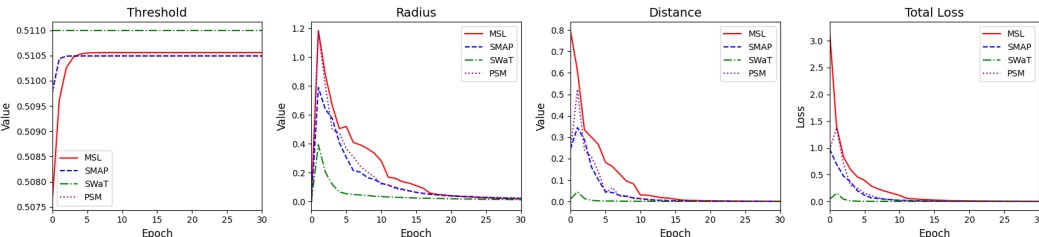

Figure 5: Visualization of the changes in threshold, radius, distance, and loss during training on four datasets.

This visualization illustrates how well the embeddings learned through each model are clustered around the center on the MSL dataset. Each model embeddings are visualized in two dimensions after training for 1, 150, and 300 epochs. All visualized data represents normal instances only, with blue points indicating correctly classified normal data and red points showing false anomaly detections. At epoch 1, both DeepSVDD and MADCluster display a dispersed distribution of data around the center. For our proposed method, 9.1%, 29.1%, and 63.3% of the data fall within 1, 2, and 3 sigma, respectively. At epoch 150, DeepSVDD exhibits a scattered distribution, while MADCluster shows data converging towards the center. MADCluster encompasses 21.0%, 51.3%, and 79.6% of the data within 1, 2, and 3 sigma, demonstrating that more data points have moved closer to the center compared to the initial epoch. By epoch 300, DeepSVDD forms multi-cluster at various points away from the center, whereas MADCluster continues to draw data closer to the center. MADCluster now includes 23.4%, 55.2%, and 82.4% of the data within 1, 2, and 3 sigma. In conclusion, as training progresses, our proposed MADCluster method shows hidden embeddings converging closer to a single cluster center, as intended. In contrast, DeepSVDD does not exhibit this tendency towards a central coordinate. Instead, it appears to form multi-cluster in the feature space, with data points grouping together with their nearby neighbors, resulting in a multi-cluster-like distribution rather than a single, centralized cluster. This visualization effectively demonstrates the enhancements over previous model constraints, addressing not only the limited expressiveness issue but also preventing the hypersphere collapse that can occur when using fixed centroids. By allowing dynamic center updates, MADCluster enables a more flexible and expressive representation of the normal data distribution in the feature space.

Figure 5 visualizes the changes in threshold, radius, distance, and loss during the training process across four datasets illustrating how each metric evolves as training progresses. The threshold, which refers to the one-directed threshold, shows a pattern of gradual increase in the early stages of training before eventually converging. After the threshold converges, radius, distance, and loss generally exhibit a decreasing trend. This pattern is consistently observed across all datasets. The proposed one-directed threshold method can serve as an indicator to assess whether the training is proceeding correctly.

## 5 CONCLUSION AND FUTURE WORK

This paper proposes a novel model-agnostic anomaly detection with self-supervised clustering network called MADCluster, which is applicable to existing deep learning anomaly detection models and addresses the hypersphere collapse problem. MADCluster consists of three modules: a Base Embedder that captures high-dimensional temporal dynamics, Cluster Distance Mapping that maps data close to normal cluster centers, and Sequence-wise Clustering that utilizes a self-learning mechanism for continuous updating of cluster centers. When applying MADCluster to comparative models across four benchmark datasets, we empirically observed that the learning of center coordinates gains more expressiveness, leading to performance improvements. Notably, MADCluster effectively improves anomaly scores by enhancing recall, though this remains an experimental observation with limitations in clearly understanding how specific structural characteristics of the model improve recall. Furthermore, as Base Embedder is only effective when it can extract temporal dynamics with sufficient expressiveness, future research should focus on developing methodologies that increase applicability not only to traditional machine learning techniques but also to deep learning models with various architectures.

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

# A PROOF OF THE ONE-DIRECTED ADAPTIVE LOSS FUNCTION

In this chapter, we will explain our own loss function. First, we analyze why Binary Cross Entropy (BCE) is inadequate for our situation. What we're trying to achieve serves as a clear motivation for a newly constructed loss function. Then, using the properties of a function whose exponent is a positive rational number less than 1, a new loss function is defined. In the last part of this chapter, the derivative of this loss function and the sign of the derivative are mathematically considered, to ensure that the total loss function actually decreases during the learning process. For simplicity in this Appendix, we will use $q$ and $p$ to represent $q_t$ and $p_t$ respectively, without loss of generality. This notation will be used consistently throughout the following proofs and explanations.

## A.1 MOTIVATION FOR PROPOSING ONE-DIRECTED ADAPTIVE LOSS

### A.1.1 ANALYSIS TO BINARY CROSS ENTROPY

We will first examine a brief analysis of the BCE. The loss function is constructed as follows:

$$\mathcal{L}_{\text{cluster}} = -\sum p \log q + (1-p) \log(1-q) \tag{8}$$

Before calculating $p$ by equation 4 using one-directed threshold, assume that the threshold is fixed as 0.5 in the loss function. Then, $p$ is determined by the following rule:

$$p = \begin{cases} 0, & 0 \leq q < 0.5 \\ 1, & 0.5 \leq q \leq 1 \end{cases} \tag{9}$$

So the loss function is calculated by different functions depending on which interval the value of $q$ belongs to. In the BCE, the total interval $[0, 1]$ for the available value of $q$ is divided by a threshold, which is 0.5, into two different intervals: $[0, 0.5)$ and $[0.5, 1]$. To simplify the analysis, let's consider a function where the variable $q$ is on the $x$-axis and the value inside the logarithm is on the $y$-axis. Then we can reconstruct the original BCE into:

$$y = \begin{cases} 1-q, & 0 \leq q < 0.5 \\ q, & 0.5 \leq q \leq 1 \end{cases} \tag{10}$$

Figure 6 shows the value inside the logarithm in the BCE loss function. To reduce the total loss, the value inside the logarithm must be increased.

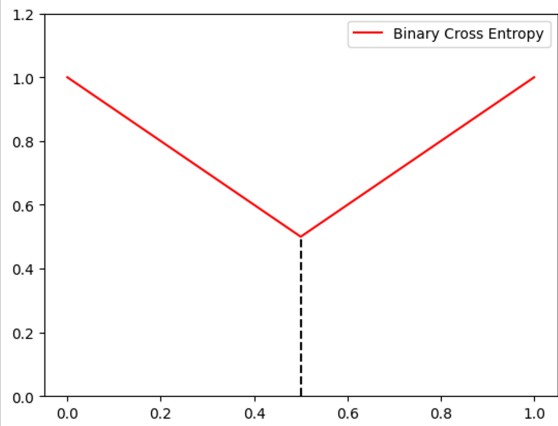

Figure 6: The black dashed line represents the position of the threshold that acts on the value $q$ to classify whether the label is 0 or 1.

Therefore, the closer the value of $y$ is to 1, the smaller the total loss. The distribution of $q$ can therefore be classified into two different labels. One will be located in the neighborhood of 0 and the other will be located in the neighborhood of 1. However, this approach poses a problem in anomaly detection tasks using single clustering, particularly when training only on normal data. The issue arises because the BCE loss function allows normal data to be correctly classified whether it's close to 0 or 1. We typically want normal data to cluster towards one direction - either 0 or 1, not both. The learning process should encourage normal data to converge towards a single value (either 0 or 1), rather than allowing it to be distributed at both extremes.

### A.1.2 DESIRED GOALS

What we are aiming for requires two differences from the original loss function. The first one is that the threshold must be learned, and the threshold must increase as it is learned. And second, the distribution of $q$ should only be close to 1, not to 0, during the learning process. If the threshold is denoted by $\nu$, we will take a monotonic function such that the overall graph should approach $y = 1$ as the value of $\nu$ increases as a value part of the logarithm of a new loss function.

### A.2 THE ONE-DIRECTED ADAPTIVE LOSS FUNCTION MODELING

At first, the total interval $[0, 1]$ in which all possible $q$ values is divided into $[0, \nu)$ and $[\nu, 1]$. Then the value $p$ is determined as follows:

$$p = \begin{cases} 0, & 0 \le q < \nu \\ 1, & \nu \le q \le 1 \end{cases} \tag{11}$$

To avoid the situation where the loss function is not defined, assume that the possible $\nu$ is in the range $0 < \nu < 1$. The simplest monotonic function connecting two points $(0,0)$ and $(1,1)$ is of the form $y = q^n$. For $n$ which satisfies the inequality $0 < n < 1$, the functions $y = q^n$ are close to $y = 1$ as $n$ decreases. So consider the following function to match the increasing trend of $\nu$ with the decreasing trend of $n$:

$$y = q^{1-\nu} \tag{12}$$

Figure 7 shows the graphs of the above function with different values of $\nu$ between 0 and 1. As $\nu$ increases, it can be seen that starting from $y = x$ and approaching $y = 1$ rapidly. This effect is more pronounced at lower values of $q$.

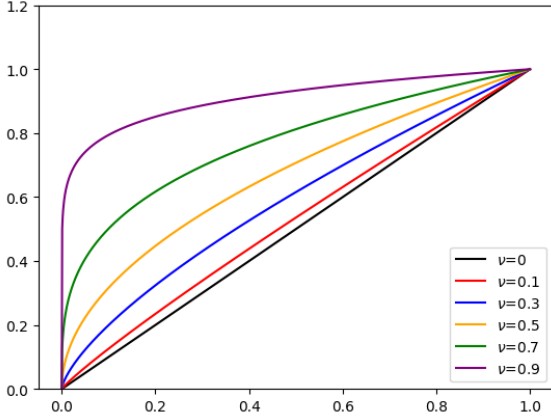

Figure 7: The graph of $y = q^{1-\nu}$ with different values of $\nu$ between 0 and 1.

Qualitatively, this function is rapidly increasing to 1 for small $q$ when $\nu$ is increasing. So we adopt the function $q^{1-\nu}$ in the interval $[0, \nu)$ as the value inside the logarithm of the loss function. Meanwhile, in the interval $[\nu, 1]$, we define the function as a linear function connecting two points $(\nu, \nu^{1-\nu})$ and $(1, 1)$, ensuring the continuity of the entire function over the interval $[0, 1]$ and reflecting the simplest form.

$$y = \frac{1 - \nu^{1-\nu}}{1 - \nu}(q - \nu) + \nu^{1-\nu} = \frac{1 - \nu^{1-\nu}}{1 - \nu}(q - 1) + 1 \tag{13}$$

In summary, we adopt the following function as the value inside the logarithm of our new loss function.

$$y = \begin{cases} q^{1-\nu}, & 0 \leq q < \nu \\ \frac{1-\nu^{1-\nu}}{1-\nu}(q - 1) + 1, & \nu \leq q \leq 1 \end{cases} \tag{14}$$

Corresponding graphs with different $\nu$ are shown in Figure 8. Each colored dashed line indicates the position of the threshold at different values of $\nu$. Before the threshold, the function is concave; after it, the function is linear.

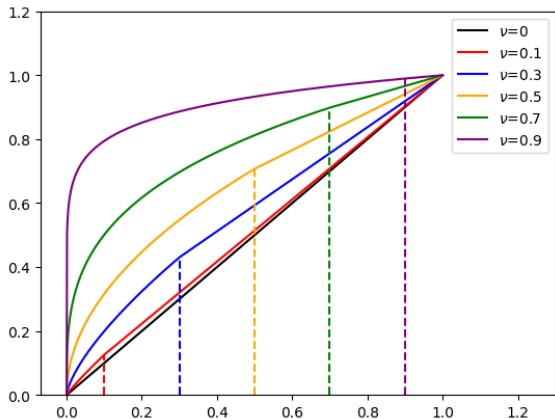

Figure 8: The graph of our new loss function with different values of $\nu$ between 0 and 1.

Thus, the final loss function can be expressed as follows:

$$\mathcal{L}_{\text{cluster}} = -\sum p \log\left(\frac{1 - \nu^{1-\nu}}{1 - \nu}(q - 1) + 1\right) + (1 - p) \log\left(q^{1-\nu}\right) \tag{15}$$

### A.3 DERIVATIVE OF LOSS FUNCTION

In order to mathematically confirm that the new loss function really decreases when $q$ and $\nu$ are increasing, to simplify the derivative procedure, let us define $f_1$ and $f_2$ as:

$$f_1 \equiv \frac{1 - \nu^{1-\nu}}{1 - \nu}(q - \nu) + \nu^{1-\nu} = \frac{1 - \nu^{1-\nu}}{1 - \nu}(q - 1) + 1, \qquad f_2 \equiv q^{1-\nu} \tag{16}$$

Since both $f_1$ and $f_2$ satisfy the conditions for a valid logarithm argument, $f_1$ and $f_2$ are positive in the entire interval $[0, 1]$. The derivative of total loss $\mathcal{L}_{\text{cluster}}$ with respect to $q$ and $\nu$ can be expressed as:

$$\frac{\partial \mathcal{L}_{\text{cluster}}}{\partial q} = -p\frac{1}{f_1}\frac{\partial f_1}{\partial q} - (1-p)\frac{1}{f_2}\frac{\partial f_2}{\partial q}, \qquad \frac{\partial \mathcal{L}_{\text{cluster}}}{\partial \nu} = -p\frac{1}{f_1}\frac{\partial f_1}{\partial \nu} - (1-p)\frac{1}{f_2}\frac{\partial f_2}{\partial \nu}. \qquad (17)$$

### A.3.1 $\partial \mathcal{L}_{\text{CLUSTER}}/\partial q$

Since both $f_1$ and $f_2$ are positive, we need to verify the signs of $\partial f_1/\partial q$ and $\partial f_2/\partial q$. Let's consider the derivative of $f_1$ with respect to $q$ first:

$$\frac{\partial f_1}{\partial q} = \frac{1 - \nu^{1-\nu}}{1 - \nu} \qquad (18)$$

The condition $0 < \nu < 1$ implies $0 < \nu^{1-\nu} < 1$. Therefore, both the denominator and the numerator are positive, ensuring that $\partial f_1/\partial q > 0$ is satisfied. Meanwhile, the derivative of $f_2$ with respect to $q$ can be written as:

$$\frac{\partial f_2}{\partial q} = (1 - \nu)q^{-\nu} = \frac{1 - \nu}{q^\nu} \qquad (19)$$

Similarly, because $0 < \nu < 1$ and $0 < q < 1$, both the denominator and the numerator are also positive, so $\partial f_2/\partial q > 0$ is satisfied. Thus, we can determine the sign of the derivative of our new loss function with respect to $q$:

$$\frac{\partial \mathcal{L}_{\text{cluster}}}{\partial q} < 0 \qquad (20)$$

This means that the total loss $\mathcal{L}_{\text{cluster}}$ decreases as $q$ increases.

### A.3.2 $\partial \mathcal{L}_{\text{CLUSTER}}/\partial \nu$

This part is very similar to proving the sign of $\partial \mathcal{L}_{\text{cluster}}/\partial q$, but it requires a more technical procedure. The derivative of total loss $\mathcal{L}_{\text{cluster}}$ with respect to $\nu$ can be written as follows:

$$\frac{\partial \mathcal{L}_{\text{cluster}}}{\partial \nu} = -p\frac{1}{f_1}\frac{\partial f_1}{\partial \nu} - (1-p)\frac{1}{f_2}\frac{\partial f_2}{\partial \nu} \qquad (21)$$

Since both $f_1$ and $f_2$ are positive, we need to verify the signs of $\partial f_1/\partial \nu$ and $\partial f_2/\partial \nu$. Let's consider the derivative of $f_1$ with respect to $\nu$ first:

$$\begin{aligned}
\frac{\partial f_1}{\partial \nu} &= \frac{(q-1)}{(1-\nu)^2}\left[-\nu^{1-\nu}\left(\frac{1-\nu}{\nu} - \log\nu\right)(1-\nu) + (1-\nu^{1-\nu})\right] \\
&= \frac{(q-1)}{(1-\nu)^2}\left[1 + \nu^{1-\nu}\left(-\frac{(1-\nu)^2}{\nu} + (1-\nu)\log\nu - 1\right)\right] \\
&= \frac{(q-1)}{(1-\nu)^2\nu^\nu}\left\{\nu^\nu + \nu - \nu^2 - 1 + \nu(1-\nu)\log\nu\right\} \qquad (22)
\end{aligned}$$

We have a condition for $c$ and $q$, which is $0 < \nu < 1$ and $0 < q < 1$. The outermost factor satisfies the following inequality:

$$\frac{(q-1)}{(1-\nu)^2 \nu^\nu} < 0 \tag{23}$$

Let us define $g_1, g_2, g_3$ as:

$$\begin{cases} g_1 = \nu^\nu + \nu \\ g_2 = \nu^2 + 1 \\ g_3 = \nu(1-\nu)\log\nu \end{cases} \tag{24}$$

To express the formula inside the braces as $g_1 - g_2 + g_3$, we will confirm the sign of each function for $\nu \in (0,1)$, thereby justifying the sign of the formula inside the braces. $g_3$ satisfies $g_3 < 0$ because of two inequalities:

$$\log\nu < 0, \qquad \nu(1-\nu) > 0 \tag{25}$$

From the limit $\lim_{\nu \to 0+} \nu^\nu = 1$, we can obtain the values of $g_1$ and $g_2$ at $\nu = 1$ and the left-side limit values of $g_1$ and $g_2$:

$$\begin{cases} g_1(0+) = g_2(0+) = 1 \\ g_1(1) = g_2(1) = 2 \end{cases} \tag{26}$$

The derivative of $g_1$ with respect to $\nu$ is:

$$\frac{\partial g_1}{\partial \nu} = \nu^\nu(1 + \log\nu) + 1 \tag{27}$$

Here, the first term $\nu^\nu(1 + \log\nu)$ is negative when $\nu \in (0, e^{-1})$, while it is positive due to the factor $(1 + \log\nu)$ when $\nu \in (e^{-1}, 1)$. Consequently, the function $g_1 - \nu$ decreases in the interval $(0, e^{-1})$ and increases in the interval $(e^{-1}, 1)$. Additionally, the first term $\nu^\nu(1 + \log\nu)$ diverges to $-\infty$ as $\nu$ approaches 0 from the positive side. While the interval of increase or decrease might differ by adding the constant 1 to the first term, the overall trend of $g_1$ remains the same even when considering $g_1 - \nu$. The derivative of $g_2$ with respect to $\nu$ is:

$$\frac{\partial g_2}{\partial \nu} = 2\nu \tag{28}$$

This quantity is always positive if $\nu \in (0,1)$, so the function $g_2$ increases in the interval $(0,1)$. Therefore, in the interval $(0,1)$, the function $g_1$ is always smaller than the function $g_2$; $g_1 - g_2 < 0$. This means that the formula $g_1 - g_2 + g_3$ satisfies the following inequality where $\nu \in (0,1)$:

$$g_1 - g_2 + g_3 < 0 \tag{29}$$

Indeed, the graph of $g_1 - g_2 + g_3$ represents negative values in the interval $(0,1)$, as shown in Figure 9.

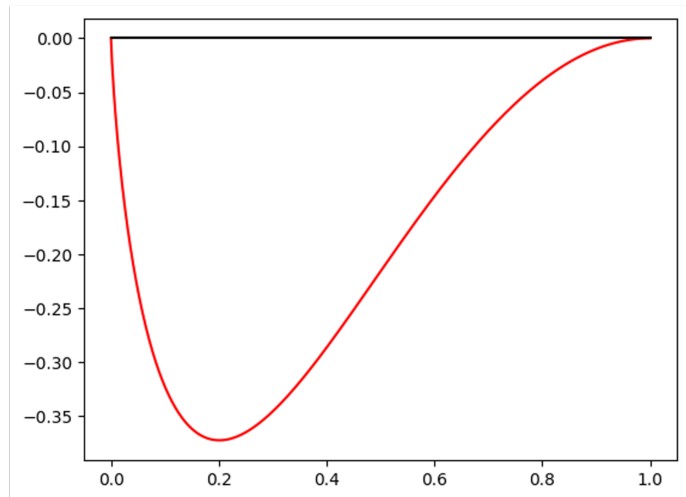

Figure 9: The graph of $g_1 - g_2 + g_3$ in the interval $[0, 1]$. The black line represents the $x$-axis; values below this line indicate that the function is negative.

Therefore, the sign of the derivative of $f_1$ with respect to $\nu$ is positive, so $\partial f_1 / \partial \nu > 0$. On the other hand, for $\partial f_2 / \partial \nu$, we have:

$$\frac{\partial f_2}{\partial \nu} = -q^{1-\nu} \log q \tag{30}$$

The value of $q^{1-\nu}$ is between 0 and 1, and $\log q < 0$, so $\partial f_2 / \partial \nu > 0$. Thus, we can determine the sign of the derivative of our new loss function with respect to $\nu$:

$$\frac{\partial \mathcal{L}_{\text{cluster}}}{\partial \nu} < 0 \tag{31}$$

This means that the total loss $\mathcal{L}_{\text{cluster}}$ decreases as $\nu$ increases.

## B  MULTI-CLUSTER ($k > 1$) FOR MADCLUSTER

MADCluster employs cosine similarity with a One-directed Adaptive loss function, initially assuming a single cluster ($k = 1$). This design overcomes the trivial solution where the soft assignment of a student's $t$-distribution always yields a value of 1 when only one cluster is present. Whereas, with several modifications, MADCluster can be extended utilizing student's $t$-distribution to support multi-cluster based clustering ($k > 1$). The soft assignment $q_{tj}$ and the target distribution $p_{tj}$ represent the assignment of the $t$-th representation to the $j$-th cluster and is defined as:

$$q_{tj} = \frac{(1 + |h_t^f - \hat{c}_j|^2)^{-1}}{\sum_{j=1}(1 + |h_t^f - \hat{c}_j|^2)^{-1}}, \qquad p_{tj} = \frac{q_{tj}^2 / \sum_{t=1} q_{tj}}{\sum_{j=1}(q_{tj}^2 / \sum_{t=1} q_{tj})} \tag{32}$$

Sequence-wise Clustering loss $\mathcal{L}_{\text{cluster}}$ is calculated using the Kullback-Leibler (KL) divergence instead of the One-directed Adaptive loss. It is defined as follows:

$$\mathcal{L}_{\text{cluster}} = KL(P|Q) = \sum_{j=1}^{K} \sum_{t=1}^{T} p_{tj} \log \frac{p_{tj}}{q_{tj}} \tag{33}$$

And for the Cluster Distance Mapping loss $\mathcal{L}_{\text{distance}}$ we have adopted a simplified notation, omitting some details for clarity, is also defined as follows:

$$\mathcal{L}_{\text{distance}} = \frac{1}{n} \sum_{j=1}^{K} \sum_{t=1}^{T} \|h_t^f - \hat{c}_j\|^2 + \lambda \Omega(\mathcal{W}) \tag{34}$$

Consequently, during training, we sum two components for each time step $t$: the KL-divergence values across all clusters for the $t$-th representation, and the distances from the $t$-th representation to each cluster center. The anomaly score is also defined as follows:

$$\text{Anomaly Score}(x_t) = \sum_{j=1}^{K} p_{tj} \log \frac{p_{tj}}{q_{tj}} + \left\| h_t^f - c_j^* \right\|^2 \tag{35}$$

For the multi-cluster case, the anomaly score does not incorporate $\nu$, and therefore $\nu$ is not learned. Similar to the single-cluster case, $\text{Anomaly Score}(x_t) \in \mathbb{R}^{T \times 1}$ serves as the point-wise anomaly score for $\mathcal{X}$.

Furthermore, we conducted experiments using multi-cluster with k=1,2,3,4,5,6,7,10. The experimental results for multi-cluster, which utilize the modified equation, are presented in Table Table 2.

Table 2: Results of evaluating MADCluster performance on four real-world datasets with multi-cluster ($k = 1$ to 10).

| Dataset | MSL | | | SMAP | | | SWaT | | | PSM | | |
|---|---|---|---|---|---|---|---|---|---|---|---|---|
| # Clusters | P | R | F1 | P | R | F1 | P | R | F1 | P | R | F1 |
| 1 | 91.83 | 98.07 | **94.84** | 93.58 | 99.36 | 96.39 | 93.24 | 100.00 | **96.50** | 97.42 | 97.94 | 97.68 |
| 2 | 91.88 | 95.74 | 93.77 | 93.61 | 99.36 | **96.40** | 92.47 | 100.00 | 96.09 | 97.47 | 98.53 | **98.00** |
| 3 | 85.89 | 95.74 | 90.55 | 93.67 | 99.25 | 96.38 | 73.44 | 100.00 | 84.69 | 97.19 | 98.76 | 97.97 |
| 4 | 87.14 | 95.74 | 91.24 | 93.57 | 97.87 | 95.67 | 50.04 | 100.00 | 66.70 | 72.23 | 98.39 | 83.31 |
| 5 | 91.88 | 95.74 | 93.77 | 93.67 | 98.88 | 96.20 | 46.17 | 100.00 | 63.17 | 70.41 | 95.63 | 81.10 |
| 6 | 84.98 | 95.74 | 90.04 | 93.16 | 98.52 | 95.76 | 58.03 | 100.00 | 73.44 | 70.83 | 96.76 | 81.78 |
| 7 | 85.10 | 98.07 | 91.13 | 93.34 | 98.08 | 95.65 | 58.83 | 100.00 | 74.08 | 72.37 | 94.00 | 81.78 |
| 8 | 85.11 | 98.07 | 91.13 | 93.22 | 93.94 | 93.58 | 43.16 | 100.00 | 60.29 | 71.67 | 94.00 | 81.33 |
| 9 | 85.09 | 98.07 | 91.12 | 93.35 | 97.13 | 95.20 | 31.93 | 100.00 | 48.40 | 78.92 | 93.94 | 85.77 |
| 10 | 85.09 | 98.07 | 91.12 | 93.83 | 96.72 | 95.26 | 12.14 | 100.00 | 21.65 | 91.34 | 93.92 | 92.61 |

Overall, across all benchmark datasets—MSL, SMAP, SWaT, and PSM—the detection performance tends to be better with one or two clusters than with a larger number of clusters. Generally, the best performance was observed with one cluster, and while the SMAP and PSM dataset showed the best performance with two clusters, this improvement was not significantly better than with one cluster and this decrease is not offset by any notable advantages. Moreover, as the number of clusters increases, performance decreases across all datasets generally. This suggests that even a small number of clusters can adequately model the normal features of complex datasets, offering advantages in terms of model interpretability when compared to scenarios with more clusters.

Consequently, in the proposed framework, the performance varies with the number of clusters, and generally, fewer clusters yield better performance. This highlights the importance as a critical element of proposed One-directed Adaptive loss function by proven on the single cluster.

# C    RESULTS AFTER APPLYING MADCLUSTER TO BASELINE MODELS

## C.1    COMPUTATIONAL EFFICIENCY

Table 3 lists the computational costs and validation accuracy, with all models trained on the MSL dataset. When applying MADCluster, performance significantly improves without substantially impacting structural complexity or efficiency. This integration results in only a slight increase in computational demands, as measured by MACs (KMac units), with a modest increase in parameter size. By maintaining a balance between efficiency and performance, this method enhances the anomaly detection capabilities of existing models without imposing significant changes. This demonstrates the effectiveness and adaptability of MADCluster, indicating its potential to improve existing anomaly detection techniques while balancing computational demands and performance enhancement.

Table 3: Computational Efficiency and F1 Score Comparison on the MSL Dataset, detailing the number of parameters ('# Params') indicating model size and Multiply-Accumulate Computations ('MACs') reflecting processing speed.

| Model | MACs | #Params | F1 |
|---|---|---|---|
| DilatedRNN | 31.81M | 311.55K | 81.24 |
| **DilatedRNN + MADCluster** | 31.81M | 311.62K | **94.84** |
| USAD | 427.36M | 256.26M | 89.13 |
| **USAD + MADCluster** | 427.36M | 256.26M | **93.72** |
| THOC | 69.42M | 390.78K | 89.69 |
| **THOC + MADCluster** | 69.42M | 390.91K | **93.76** |
| AnoTrans | 485.23M | 4.86M | 93.93 |
| **AnoTrans + MADCluster** | 485.23M | 4.86M | **94.90** |
| DCdetector | 1189.00M | 912.18K | 94.79 |
| **DCdetector + MADCluster** | 1189.00M | 912.30K | **95.18** |

## C.2    COMPARISON OF ANOMALY DETECTION APPROACHES

In Table 4 we evaluated the performance of the anomaly detection approaches illustrated in maintext Figure 2. This table presents quantitative results of our proposed method, which learns center coordinates and performs single clustering as we hypothesized. DeepSVDD represents only distance mapping, while Clustering denotes the experimental results using self-labeling without distance mapping. MADCluster, our proposed method, applies both distance mapping and clustering.

Table 4: Performance comparison of anomaly detection approaches across four datasets: (1) DeepSVDD (Cluster Distance Mapping), (2) Clustering (Sequence-wise Clustering), and (3) MAD-Cluster (Combined Cluster Distance Mapping and Sequence-wise Clustering)

| DATASET | MSL | | | SMAP | | | SWAT | | | PSM | | |
|---|---|---|---|---|---|---|---|---|---|---|---|---|
| METRIC | P | R | F1 | P | R | F1 | P | R | F1 | P | R | F1 |
| DEEPSVDD | 88.88 | 74.81 | 81.24 | 93.58 | 99.29 | 96.35 | 89.80 | 100.00 | 94.63 | 97.59 | 96.52 | 97.05 |
| CLUSTERING | 98.95 | 49.91 | 66.35 | 93.37 | 95.84 | 94.59 | 93.17 | 100.00 | 96.47 | 99.38 | 22.32 | 36.45 |
| MADCLUSTER | 91.83 | 98.07 | **94.84** | 93.58 | 99.36 | **96.39** | 94.40 | 100.00 | **97.12** | 97.42 | 97.94 | **97.68** |

Comparing DeepSVDD and Clustering alone, DeepSVDD generally demonstrates better performance. Particularly in MSL, SMAP, and PSM datasets, DeepSVDD outperforms due to Clustering lower recall. However, Clustering shows superior results in the SWaT dataset. MADCluster, which improves upon both methods, consistently achieves the highest F1-scores across all models and datasets. This indicates that MADCluster effectively enhances the balance between precision and recall, thereby strengthening anomaly detection capabilities.

### C.3 PARAMETER SENSITIVITY

We conducted experiments to assess the sensitivity of our proposed model performance to various parameters. Figure 10 illustrates the results across all four datasets for the following parameter ranges: window sizes (25, 50, 75, 100, 125, 150, 175, 200), number of clusters (1-10), smoothing factors (0.0-0.5 in 0.1 increments), and thresholds (0.1-0.9 in 0.1 increments). All experiments used the dilated RNN model with MADCluster applied.

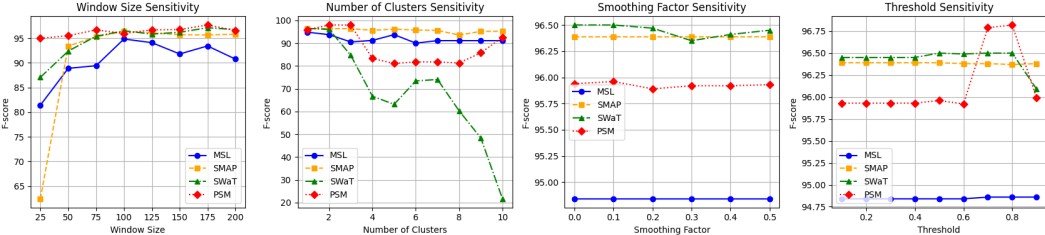

Figure 10: Performance sensitivity to window size variations across various datasets. This illustrates the importance of selecting an optimal window size based on the specific data patterns of each dataset.

Our findings show that performance generally improved with larger window sizes across all datasets, except for MSL, where smaller window sizes consistently yielded lower performance. Lower cluster numbers generally produced better overall performance, with the SWaT dataset notably exhibiting a dramatic performance degradation as the number of clusters increased. While performance variation was relatively small across different smoothing factors, lower values tended to yield the best results. For the threshold parameter, we observed similar trends to the smoothing factor up to 0.5. However, beyond 0.6, we observed increased performance for the PSM dataset, while the SWaT dataset experienced a sharp performance drop at a threshold of 0.9. These results provide valuable insights into the optimal parameter settings for our proposed model across different datasets and highlight the importance of careful parameter tuning in anomaly detection tasks.

## D  DATASET

We summarize the four adopted benchmark datasets for evaluation in Table 5. These datasets include multivariate time series scenarios with different types and anomaly ratios. MSL, SMAP, SWaT and PSM are multivariate time series datasets.

Table 5: Statistics and details of the benchmark datasets used. AR (anomaly ratio) represents the abnormal proportion of the whole dataset.

| BENCHMARKS | APPLICATIONS | DIM | WIN | #TRAIN | #TEST | AR (TRUTH) |
|---|---|---|---|---|---|---|
| MSL | SPACE | 55 | 100 | 58,317 | 73,729 | 0.105 |
| SMAP | SPACE | 25 | 100 | 135,183 | 427,617 | 0.128 |
| SWAT | WATER | 51 | 100 | 495,000 | 449,919 | 0.121 |
| PSM | SERVER | 25 | 100 | 132,481 | 87,841 | 0.278 |

