# OpenReview forum: "MADCluster: Model-agnostic Anomaly Detection with Self-supervised Clustering Network"
_ICLR.cc/2025/Conference — Submitted to ICLR 2025_

### Official Review · Reviewer_VmN2 · 2024-11-01

**Soundness:** 3
**Presentation:** 4
**Contribution:** 2
**Rating:** 5
**Confidence:** 3

**Summary:**

This paper introduces a model-agnostic anomaly detection approach that incorporates a novel self-supervised clustering step to mitigate hypersphere collapse issues in one-class classification methods. The approach leverages a sequence-wise clustering mechanism that updates the central coordinates of the hypersphere via network parameters, using a one-directional adaptive loss function. The authors provide a mathematical formulation and analysis of the optimization process, showing that it achieves the intended effect. Experimental results on four benchmark datasets and comparisons with 11 baseline methods validate the approach’s effectiveness.

**Strengths:**

1.	Innovative Loss Function: The paper introduces a novel one-directional adaptive loss function designed to enable distance-based mapping through parameter updates. The authors carefully derive and discuss the impact of this mapping, demonstrating its ability to capture coherent anomalies in similar regions and reduce inconsistent labeling among similar data points. The derivation is thorough and mathematically well-supported, enhancing the approach’s theoretical soundness.
	2.	Extensive Comparative Analysis: The method’s model-agnostic capability is demonstrated by applying it across a variety of deep learning models and comparing results with over ten baselines. The qualitative results indicate that the proposed method learns a more expressive and meaningful representation for hypersphere construction, which strengthens its potential for effective anomaly detection.

**Weaknesses:**

1.	Limited Innovation Scope: The paper aims to address hypersphere collapse, which is already a well-explored issue in a one-class classification problem (DASVDD, THOC). The clustering-based approach itself is not novel as well (DEC). Although the one-directional adaptive loss function represents an interesting advancement, the extent to which it genuinely enhances anomaly detection is unclear. For example, from Figure 2 it's hard to tell if there is an improved performance in (c) over (b) when distance mapping is employed, as the performance largely depends on the definition of the anomaly.
	2.	Marginal Performance Gains: Table 1 shows consistent improvements for MADCluster over baselines across datasets, but the gains appear marginal (if we compare the best performer with the second best one, for example) and might lack statistical significance. Including standard deviations for these results would help validate the claims of performance improvement, providing a more robust basis for assessing the method’s advantages.
	3.	Need for Ablation Study: An ablation study on the F1 scores across different datasets, specifically isolating the contributions of clustering and distance mapping, would offer stronger evidence of each component’s effectiveness. Relying solely on the illustration in Figure 2 limits the understanding of their respective roles in the overall framework.

**Questions:**

•	Can you clarify how your approach uniquely addresses hypersphere collapse beyond existing methods? Specifically, what is the key contribution of your one-directional adaptive loss function compared to prior approaches?
	•	How do you define an “anomaly” within the context of your method? Can you provide a clearer explanation of how your distance-based mapping approach influences anomaly detection outcomes?
	•	In Figure 2, it’s challenging to see a distinct improvement from adding distance mapping in subfigure (c) versus (b). Can you provide additional analysis or quantitative comparisons to clarify how the mapping contributes to anomaly coherence and detection accuracy?

---

> ### Author Response · Authors · 2024-11-22
>
> We sincerely thank the reviewer VmN2 for this thoughtful question.
>
> **Q1 & Q3 & W1 & W3**
> > **Regarding the summary of questions: “Limited Innovation and Novelty”**
>
>
> * The proposed one-directed adaptive loss is theoretically supported to perform better in classifying anomalies.
>
>   * **Criteria for distinguishing normal data and anomaly data (Cosine similarity based)**
>     * We calculated the cosine similarity $q_t$ between the embedded $h^f_t$ in the feature space and the cluster center coordinate $\hat{c}$ using **equation (3)**. Then we applied the transformation $(q_t+1)/2$, adjusting the range of $q_t$ to lie between 0 and 1.
>     * This approach means that as $q_t$ approaches 1, the data representation on feature space aligns more closely with the normal cluster center. Consequently, when $q_t$ is near 1, the data can be classified as normal, whereas values closer to 0 indicate an anomalous classification. Specifically, for the two discrete labels 0 and 1, values of $q_t$ above a defined threshold $\nu$ are labeled as normal, while values below this threshold indicate anomaly.
>
>   * **Adapting Thresholds During Normal Data Training**
>     * Meanwhile, the objective of the distance loss **equation (2)** is to enclose a significant amount of normal data while minimizing the hypersphere radius $R$. As the learning process of the distance loss progresses, the criteria for classifying data as normal become more stringent, meaning the threshold $\nu$ in **equation (4)** should be gradually increased. Since our method learns only from normal data, both the data labeled as 1 and the increasing of threshold $\nu$ in the one-directed adaptive loss **equation (5)** have to adapt accordingly as learning continues.
>
>   * **Summary and application in anomaly detection**
>     * In summary, from a theoretical perspective, normal data distribution becomes more concentrated around the cluster center coordinates $\hat{c}$ under one-directed adaptive loss **equation (5)**, compared to models without this loss function.
>     * We assume that the value of $q_t$ corresponding to the anomaly data will be significantly less than 1 when new anomaly data comes in. This result in the learning approach effectively distancing anomalous data outside the hypersphere, thus aiding in anomaly detection. We think **Figure 4** sufficiently reflects this trend, even though it's qualitative.
>
> * Ultimately, the proposed one-directed adaptive loss functions to ensure that normal data becomes more densely concentrated around the cluster center coordinate $\hat{c}$.
>
> > **Regarding the summary of questions: “Need for Comprehensive Ablation Study”**
>
> * This outcome is reflected in the experimental results presented in **Table 4** of **Appendix C.2.** In **Table 4**, the results for DeepSVDD represent the use of $\mathcal{L}\_{\text{distance}}$ with fixed center coordinates only, while the results for Clustering reflect the use of $\mathcal{L}\_{\text{cluster}}$, which is a one-directed adaptive loss, without employing $\mathcal{L}\_{\text{distance}}$. Finally, MADCLUSTER incorporates both $\mathcal{L}\_{\text{distance}}$ and $\mathcal{L}\_{\text{cluster}}$.
>
>
> **Q2**
> > **How do you define an “anomaly” within the context of your method? Can you provide a clearer explanation of how your distance-based mapping approach influences anomaly detection outcomes?**
>
>
> * The proposed MADCluster method is designed as a model-agnostic anomaly detection framework, which also addresses the issue of ‘hypersphere collapse’ seen in prior methods [1]. Given its flexibility and adaptability across domains, the specific definition of “anomaly” can vary depending on the application.
>
>     [1] Deep One-Class Classification. PMLR, 2018.
>
>
> - Additionally, the **Cluster distance Mapping** approach operates by calculating the distance between the cluster center and the mapped points in the latent space. This ensures that anomalies are identified based on their deviation from the normal data distribution within this space. Because the detection relies on these latent representations rather than explicit anomaly definitions, we believe that it is difficult to consider that the definition of outliers significantly impacts performance.

---

> > ### Comment · Reviewer_VmN2 · 2024-11-25
> >
> > The author’s response addresses most of my questions. However, I remain unconvinced by the answer to Q2. Based on my experience with anomaly detection in industrial tasks, it is crucial to identify the specific types of anomalies a method is best suited to detect. While the author claims that the approach ensures “anomalies are identified based on their deviation from the normal data distribution within this space,” allowing it to handle different types of anomalies, Figure 2 suggests otherwise. It appears that the method may exhibit a bias towards detecting specific types of anomalies, such as data shifting to one side of the distribution.

---

> > > ### Author Response · Authors · 2024-11-26
> > >
> > > Dear Reviewer VmN2, thank you for your valuable Question 2.
> > >
> > >
> > > * It appears that your concerns primarily focus on the issues of point-wise outliers and pattern-wise outliers (e.g., on global, contextual, shapelet, seasonal, and trend anomalies), as discussed in [1]. Indeed, identifying and addressing various types of anomalies is crucial in both industrial and real-world applications.
> > >
> > >
> > >
> > > * We agree that these anomalies often fall under the categories outlined in [1]. To better understand and address such anomalies, [2, 3] conducted ablation studies, which demonstrated the ability to detect diverse types of anomalies. These studies highlight the significance of a robust framework capable of accommodating multiple anomaly types.
> > >
> > >
> > >
> > > * Our proposed model, as mentioned, possesses model-agnostic characteristics. This enables it to work seamlessly alongside anomaly detection methods capable of addressing diverse cases, such as those described in [2, 3]. Moreover, by incorporating the MADCluster module into these models, we can further enhance their effectiveness.
> > >
> > >
> > >     [1] Revisiting Time Series Outlier Detection: Definitions and Benchmarks. NeurIPS, 2021
> > >
> > >
> > >     [2] Anomaly Transformer: Time Series Anomaly Detection with Association Discrepancy. ICLR, 2022
> > >
> > >
> > >     [3] DCdetector: Dual Attention Contrastive Representation Learning for Time Series Anomaly Detection. KDD, 2023
> > >
> > >
> > > We hope this explanation addresses your concerns effectively. Please let us know if additional clarification or further discussion is needed. Once again, thank you for your constructive questions.

---

> > > > ### Comment · Reviewer_VmN2 · 2024-11-26
> > > >
> > > > Thank you for providing the additional clarification. The overall soundness of your work and its presentation looks good to me, and I appreciate the effort you’ve put into addressing my earlier questions. However, my remaining concern about raising the score from 5 to 6 is about the relatively marginal empirical improvement and the scope of innovation, as noted in my original comments. As such, I will maintain the overall score given in my earlier review. That said, I am not entirely certain and will leave the final decision to the Area Chair.

---

### Official Review · Reviewer_wgG7 · 2024-11-03

**Soundness:** 2
**Presentation:** 2
**Contribution:** 3
**Rating:** 5
**Confidence:** 4

**Summary:**

This paper proposes MADCluster, a model-agnostic framework for anomaly detection that addresses the "hypersphere collapse" issue common in deep anomaly detection methods. MADCluster introduces self-supervised clustering to adaptively learn the cluster center, enabling it to capture normal data patterns in a centralized latent space, thus highlighting deviations as anomalies. The authors present a novel One-directed Adaptive Loss to support clustering into a single cluster, ensuring high intra-cluster similarity for normal data. By combining the Base Embedder, Cluster Distance Mapping, and Sequence-wise Clustering modules, MADCluster achieves adaptability across diverse neural network architectures.

**Strengths:**

1. The design of MADCluster allows integration with various deep learning models, making it broadly applicable across different network architectures. This adaptability offers potential practical value, as MADCluster may serve as a plug-and-play solution in different anomaly detection contexts.

2. The authors evaluate MADCluster on multiple real-world datasets, demonstrating consistent improvements across diverse models in terms of different metrics.

3. The authors release the source-code, which improves the reproducibility.

**Weaknesses:**

1. MADCluster avoids the “hypersphere collapse” problem of traditional one-classification methods by dynamically updating the clustering centers of normal data. However, the reviewer is concerned that when the distribution of normal data is complex or there are multiple local clusters, a single dynamic center may not be sufficient to accurately describe the normal pattern, which may lead to boundary-blurring or false detection. Can the authors discuss this issue?

2. The one-directed adaptive loss proposed in this work is centered on the idea of gradually increasing the threshold to cluster normal data into single clusters, but it has not yet been fully verified whether this design is effective in distinguishing anomalous samples that slightly deviate from the normal clusters. No quantitative or ablative analysis of the performance of this loss function on the model is provided in the paper.

3. In the experimental part, MADCluster is mainly compared with methods such as D-RNN and USAD, but these baseline models do not cover the latest self-supervised or GAN-based anomaly detection methods, which may outperform the traditional methods on multimodal time series data. To ensure the comprehensiveness of the experimental results, it is recommended to introduce the latest baseline models to support the effectiveness of MADCluster.

4. The title of this paper seems to suggest that MADCluster can broadly be applied across various data types. However, the experimental validation is limited to time-series datasets, with no testing on other data types (e.g., structured tabular data, graphs, images). This creates a discrepancy between the title’s generality and the paper’s scope. To better align the title with the paper’s content,
 the reviewer recommends either specifying the application to time-series data in the title or adding experiments on diverse data types to substantiate the claim of model-agnostic applicability.

**Questions:**

See weakness.

---

> ### Author Response · Authors · 2024-11-22
>
> We sincerely thank the reviewer wgG7 for raising this important point and providing a valuable suggestion.
>
> **W1**
> > **MADCluster avoids the “hypersphere collapse” problem of traditional one-classification methods by dynamically updating the clustering centers of normal data. However, the reviewer is concerned that when the distribution of normal data is complex or there are multiple local clusters, a single dynamic center may not be sufficient to accurately describe the normal pattern, which may lead to boundary-blurring or false detection. Can the authors discuss this issue?**
>
> * To address the potential limitation of using a single cluster to represent complex data distributions (with multiple clusters), we conducted experiments on multi-cluster scenarios, as detailed in **Appendix B**. **Table 2** presents the results of experiments based on the D-RNN model, where multi-cluster configurations $(k = 1 \ \text{to}\ 10 )$ were tested over 15 epochs.
>
>
> * From **Table 2**, we observed that the optimal number of clusters across datasets: for the MSL and SWaT datasets, performance was highest with a single cluster $(k = 1)$, while for the SMAP and PSM datasets, the best performance was achieved with two clusters $(k = 2)$. Across all datasets, as the number of clusters increased, performance generally declined.
>
>
> * As the reviewer noted, a single cluster may initially appear insufficient to capture complex data distributions. However, we hypothesize that with extended training beyond 15 epochs, the performance degradation observed with an increased number of clusters may suggest that a single cluster could eventually be sufficient to represent complex data. This is supported by the results shown in **Figure 4** in **Section 4.4**, which visually demonstrate that, as training progresses, the data gradually converges around a single cluster. This convergence highlights the model ability to capture complex data distributions over time.
>
>
> * We hope this additional explanation addresses the reviewer’s concerns, and we are grateful for the opportunity to further elaborate on this aspect of our work.

---

> ### Author Response · Authors · 2024-11-22
>
> **W2**
> > **The one-directed adaptive loss proposed in this work is centered on the idea of gradually increasing the threshold to cluster normal data into single clusters, but it has not yet been fully verified whether this design is effective in distinguishing anomalous samples that slightly deviate from the normal clusters. No quantitative or ablative analysis of the performance of this loss function on the model is provided in the paper**
>
>
> * To classify anomalous samples that slightly deviate from the normal clusters, the parameter $\rho$ specified in **equation (2)** of the paper plays a critical role. $\rho$ serves as a control mechanism to determine the tolerance for minor deviations from the normal clusters. As such, the effectiveness of distinguishing these slightly anomalous samples is highly influenced by the choice of the $\rho$ hyperparameter.
>
>
> * However, even though the influence of $\rho$ is significant, the proposed one-directed adaptive loss is theoretically supported to perform better in classifying anomalies.
>
>   * **Criteria for distinguishing normal data and anomaly data (Cosine similarity based)**
>     * We calculated the cosine similarity $q_t$ between the embedded $h^f_t$ in the feature space and the cluster center coordinate $\hat{c}$ using **equation (3)**. Then we applied the transformation $(q_t+1)/2$, adjusting the range of $q_t$ to lie between 0 and 1.
>     * This approach means that as $q_t$ approaches 1, the data representation on feature space aligns more closely with the normal cluster center. Consequently, when $q_t$ is near 1, the data can be classified as normal, whereas values closer to 0 indicate an anomalous classification. Specifically, for the two discrete labels 0 and 1, values of $q_t$ above a defined threshold $\nu$ are labeled as normal, while values below this threshold indicate anomaly.
>
>   * **Adapting Thresholds During Normal Data Training**
>     * Meanwhile, the objective of the distance loss **equation (2)** is to enclose a significant amount of normal data while minimizing the hypersphere radius $R$. As the learning process of the distance loss progresses, the criteria for classifying data as normal become more stringent, meaning the threshold $\nu$ in **equation (4)** should be gradually increased. Since our method learns only from normal data, both the data labeled as 1 and the increasing of threshold $\nu$ in the one-directed adaptive loss **equation (5)** have to adapt accordingly as learning continues.
>
>   * **Summary and application in anomaly detection**
>     * In summary, from a theoretical perspective, normal data distribution becomes more concentrated around the cluster center coordinates $\hat{c}$ under one-directed adaptive loss **equation (5)**, compared to models without this loss function.
>     * We assume that the value of $q_t$ corresponding to the anomaly data will be significantly less than 1 when new anomaly data comes in. This result in the learning approach effectively distancing anomalous data outside the hypersphere, thus aiding in anomaly detection. We think **Figure 4** sufficiently reflects this trend, even though it's qualitative.
>
> * Ultimately, the proposed one-directed adaptive loss functions to ensure that normal data becomes more densely concentrated around the cluster center coordinate $\hat{c}$.
>
> * This outcome is reflected in the experimental results presented in **Table 4** of **Appendix C.2.** In **Table 4**, the results for DeepSVDD represent the use of $\mathcal{L}\_{\text{distance}}$ with fixed center coordinates only, while the results for Clustering reflect the use of $\mathcal{L}\_{\text{cluster}}$, which is a one-directed adaptive loss, without employing $\mathcal{L}\_{\text{distance}}$. Finally, MADCLUSTER incorporates both $\mathcal{L}\_{\text{distance}}$ and $\mathcal{L}\_{\text{cluster}}$.

---

> ### Author Response · Authors · 2024-11-22
>
> **W3**
> > **In the experimental part, MADCluster is mainly compared with methods such as D-RNN and USAD, but these baseline models do not cover the latest self-supervised or GAN-based anomaly detection methods, which may outperform the traditional methods on multimodal time series data. To ensure the comprehensiveness of the experimental results, it is recommended to introduce the latest baseline models to support the effectiveness of MADCluster.**
>
> * In fact, to address the concern regarding the inclusion of the latest self-supervised or GAN-based anomaly detection methods, we had conducted experiments referring to the results of [1], which achieved state-of-the-art performance in 2023.
>
>
> * However, we observed that the performance of these models [2,3,4], when reproduced in our experiments, did not reach the level reported in the referring paper. In fact, the results we obtained were significantly lower than those claimed. Due to these discrepancies, we decided not to include the results of [2,3,4] in our paper, as they could introduce confusion.
>
>
> * We appreciate the reviewer **wgG7**'s recommendation and will consider including additional baseline models and discussions in future work to further enhance the comprehensiveness of the experimental results.
>
>
> [1] DCdetector: Dual Attention Contrastive Representation Learning for Time Series Anomaly Detection. KDD, 2023
>
> [2] BeatGAN: Anomalous Rhythm Detection using Adversarially Generated Time Series. IJCAI, 2019
>
> [3] Robust Anomaly Detection for Multivariate Time Series through Stochastic Recurrent Neural Network. KDD, 2019
>
> [4] Multivariate Time Series Anomaly Detection and Interpretation using Hierarchical Inter-Metric and Temporal Embedding. KDD, 2021
>
>
> **W4**
> > **The title of this paper seems to suggest that MADCluster can broadly be applied across various data types. However, the experimental validation is limited to time-series datasets, with no testing on other data types (e.g., structured tabular data, graphs, images). This creates a discrepancy between the title’s generality and the paper’s scope. To better align the title with the paper’s content, the reviewer recommends either specifying the application to time-series data in the title or adding experiments on diverse data types to substantiate the claim of model-agnostic applicability.**
>
>   We sincerely thank the reviewer wgG7 for this insightful recommendation regarding the generalizability of MADCluster to other data domains.
>
> * In our implementation, the extracted dynamics $h_t^{f}$ obtained through the Base Embedder are represented as **[batch, sequence, hidden_dim]** in the code. For image data, where inputs are typically represented as **[batch_size, channels, height, width]**, it is possible to apply MADCluster by flattening the height and width dimensions **[batch_size, channels, height $\times$ width]**. This adjustment allows for the application of MADCluster to image data.
>
> * Additionally, we have conducted additional experiments applying MADCluster to anomaly detection in image data. Specifically, we utilized methods [1,2,3] and performed experiments on the MVTec AD dataset [4]. These experiments were simple evaluations conducted over 10 epochs, and the results are presented in two tables: one summarizing Image AUROC and another summarizing Pixel AUROC.
>
>     [1] Anomaly Detection via Reverse Distillation from One-Class Embedding. CVPR, 2022.
>
>     [2] PyramidFlow: High-Resolution Defect Contrastive Localization Using Pyramid Normalizing Flow. CVPR, 2023.
>
>     [3] RealNet: A Feature Selection Network with Realistic Synthetic Anomaly for Anomaly Detection. CVPR, 2024.
>
>     [4] MVTec AD — A Comprehensive Real-World Dataset for Unsupervised Anomaly Detection. CVPR, 2019

---

> > ### Author Response · Authors · 2024-11-22
> >
> > ### Image AUROC
> > | **Class Name** | **RealNet** | **RealNet+MADCluster** | **RD4AD** | **RD4AD+MADCluster** | **PyramidFlow** | **PyramidFlow+MADCluster** |
> > |:----------------:|:-------------:|:------------------------:|:-----------:|:-----------------------:|:-----------------:|:----------------------------:|
> > | **Bottle**     | 0.918       | 0.961                 | 0.991     | 0.998                | 0.778           | 0.993                     |
> > | **Cable**      | 0.631       | 0.669                 | 0.945     | 0.946                | 0.638           | 0.695                     |
> > | **Capsule**    | 0.694       | 0.698                 | 0.868     | 0.870                | 0.870           | 0.916                     |
> > | **Carpet**     | 0.969       | 0.977                 | 0.996     | 0.997                | 0.938           | 0.964                     |
> > | **Grid**       | 0.872       | 0.875                 | 0.921     | 0.945                | 0.794           | 0.824                     |
> > | **Hazelnut**   | 0.972       | 0.994                 | 1.000     | 1.000                | 0.930           | 0.935                     |
> > | **Leather**    | 0.806       | 0.830                 | 1.000     | 1.000                | 0.993           | 0.999                     |
> > | **Metal Nut**  | 0.670       | 0.688                 | 0.995     | 0.996                | 0.735           | 0.742                     |
> > | **Pill**       | 0.823       | 0.844                 | 0.936     | 0.956                | 0.810           | 0.834                     |
> > | **Screw**      | 0.552       | 0.572                 | 0.829     | 0.848                | 0.595           | 0.752                     |
> > | **Tile**       | 0.972       | 0.981                 | 0.993     | 0.994                | 0.994           | 0.995                     |
> > | **Toothbrush** | 0.553       | 0.644                 | 0.997     | 1.000                | 0.944           | 0.947                     |
> > | **Transistor** | 0.659       | 0.660                 | 0.967     | 0.970                | 0.908           | 0.936                     |
> > | **Wood**       | 0.959       | 0.966                 | 0.990     | 0.993                | 0.991           | 0.996                     |
> > | **Zipper**     | 0.882       | 0.901                 | 0.871     | 0.889                | 0.938           | 0.938                     |
> >
> > ---
> >
> > ### Pixel AUROC
> > | **Class Name** | **RealNet** | **RealNet+MADCluster** | **RD4AD** | **RD4AD+MADCluster** | **PyramidFlow** | **PyramidFlow+MADCluster** |
> > |:----------------:|:-------------:|:------------------------:|:-----------:|:-----------------------:|:-----------------:|:----------------------------:|
> > | **Bottle**     | 0.949       | 0.963                 | 0.982     | 0.986                | 0.960           | 0.974                     |
> > | **Cable**      | 0.631       | 0.897                 | 0.977     | 0.977                | 0.895           | 0.912                     |
> > | **Capsule**    | 0.901       | 0.927                 | 0.981     | 0.982                | 0.977           | 0.980                     |
> > | **Carpet**     | 0.970       | 0.984                 | 0.992     | 0.992                | 0.964           | 0.978                     |
> > | **Grid**       | 0.873       | 0.894                 | 0.942     | 0.963                | 0.941           | 0.948                     |
> > | **Hazelnut**   | 0.925       | 0.956                 | 0.991     | 0.991                | 0.964           | 0.973                     |
> > | **Leather**    | 0.968       | 0.971                 | 0.994     | 0.994                | 0.985           | 0.987                     |
> > | **Metal Nut**  | 0.754       | 0.770                 | 0.969     | 0.974                | 0.938           | 0.959                     |
> > | **Pill**       | 0.942       | 0.943                 | 0.967     | 0.968                | 0.943           | 0.956                     |
> > | **Screw**      | 0.929       | 0.946                 | 0.985     | 0.986                | 0.898           | 0.903                     |
> > | **Tile**       | 0.930       | 0.937                 | 0.953     | 0.953                | 0.962           | 0.973                     |
> > | **Toothbrush** | 0.918       | 0.924                 | 0.987     | 0.988                | 0.975           | 0.977                     |
> > | **Transistor** | 0.704       | 0.722                 | 0.890     | 0.890                | 0.965           | 0.972                     |
> > | **Wood**       | 0.930       | 0.932                 | 0.955     | 0.955                | 0.957           | 0.960                     |
> > | **Zipper**     | 0.951       | 0.962                 | 0.968     | 0.970                | 0.968           | 0.968                     |
> >
> >
> > We believe this demonstrates the adaptability of MADCluster and its potential to generalize to other domains like image.

---

> > > ### Comment · Reviewer_wgG7 · 2024-11-27
> > > **Follow-up feedback to authors**
> > >
> > > 1. I appreciate the authors for addressing my concerns (W1, W2, W4). However, my concern about W3 has not been addressed yet. Since MADCluster is a self-supervised AD method, the authors should at least demonstrate its superiority over the latest self-supervised AD methods.
> > > 2. Besides, Reviewer 3EUZ mentioned the "PA" issue, I am wondering if the authors used the same strategy in their evaluation of other baselines? From a personal perspective, I think the "PA" is indeed a somewhat controversial operation. Therefore I suggest the authors show the performance without "PA".
> > >
> > > Overall, I would like to maintain the current rating of this work due to the existing concerns.

---

### Official Review · Reviewer_Gomd · 2024-11-03

**Soundness:** 3
**Presentation:** 3
**Contribution:** 2
**Rating:** 6
**Confidence:** 3

**Summary:**

The paper introduces MADCluster, a model-agnostic anomaly detection framework that uses self-supervised clustering. It addresses the "hypersphere collapse" issue in deep learning-based anomaly detection methods by dynamically learning cluster centers to group normal patterns in a single cluster. Experiments on time-series datasets demonstrate that MADCluster enhances the performance of various baseline models, showcasing its flexibility and effectiveness in anomaly detection.

**Strengths:**

1.MADCluster is easy to adapt to a wide range of neural network architectures and effectively enhances the performance of base models.

2.The framework's dynamic update of cluster centers prevents hypersphere collapse, ensuring a more expressive feature space.

3.The writing of this paper is clear and well-structured.

**Weaknesses:**

1.The paper lacks a detailed analysis of time and computational costs, particularly a comparison between the original base model and the model combined with MADCluster.

2.The performance improvement observed in advanced methods like DCdetector appears to be less than in simpler methods such as D-RNN. Does this suggest a limitation in the effectiveness of MADCluster when feature quality is already high?

3.It appears that MADCluster is not specifically designed for time-series anomaly detection. Have you explored the generalizability of this method to other domains, such as tabular data or image data?

**Questions:**

See weakness above.

---

> ### Author Response · Authors · 2024-11-22
>
> We sincerely thank the reviewer Gomd for raising this important point.
>
> **W1**
> > **The paper lacks a detailed analysis of time and computational costs, particularly a comparison between the original base model and the model combined with MADCluster.**
>
> * We would like to kindly point out that **Table 3** in **Appendix C.1** presents the computational costs and validation accuracy for five models with MADCluster applied.
>
> * When applying MADCluster, there is only a minimal increase in the number of parameters, and the computational costs remain almost unchanged. The slight increase in parameters and computational costs is offset by the meaningful improvement in validation accuracy, as shown in the results of **Table 3**. We believe this trade-off highlights the efficiency and practicality of MADCluster, even when integrated into existing models.
>
> **W2**
> > **The performance improvement observed in advanced methods like DCdetector appears to be less than in simpler methods such as D-RNN. Does this suggest a limitation in the effectiveness of MADCluster when feature quality is already high?**
>
> * The performance improvement achieved by any method is influenced by the inherent performance ceiling of the dataset within a specific domain. For simpler methods like D-RNN, which initially have lower performance, the observed improvement appears larger as there is more room for enhancement. In contrast, advanced methods like DCdetector already exhibit high baseline performance, leaving less room for further improvement.
>
> * That said, it is important to highlight that even with smaller performance gains, MADCluster still outperforms advanced models like DCdetector, achieving state-of-the-art performance. This underscores the effectiveness of MADCluster in pushing the boundaries of performance, even when working with high-quality features in advanced models.
>
> **W3**
> > **It appears that MADCluster is not specifically designed for time-series anomaly detection. Have you explored the generalizability of this method to other domains, such as tabular data or image data?**
>
> We sincerely thank the reviewer Gomd for this insightful question regarding the generalizability of MADCluster to other data domains.
>
>
> * In our implementation, the extracted dynamics $h_t^{f}$ obtained through the Base Embedder are represented as **[batch, sequence, hidden_dim]** in the code. For image data, where inputs are typically represented as **[batch_size, channels, height, width]**, it is possible to apply MADCluster by flattening the height and width dimensions **[batch_size, channels, height $\times$ width]**. This adjustment allows for the application of MADCluster to image data.
>
> * Additionally, we have conducted additional experiments applying MADCluster to anomaly detection in image data. Specifically, we utilized methods [1,2,3] and performed experiments on the MVTec AD dataset [4]. These experiments were simple evaluations conducted over 10 epochs, and the results are presented in two tables: one summarizing Image AUROC and another summarizing Pixel AUROC.
>
>     [1] Anomaly Detection via Reverse Distillation from One-Class Embedding. CVPR, 2022.
>
>     [2] PyramidFlow: High-Resolution Defect Contrastive Localization Using Pyramid Normalizing Flow. CVPR, 2023.
>
>     [3] RealNet: A Feature Selection Network with Realistic Synthetic Anomaly for Anomaly Detection. CVPR, 2024.
>
>     [4] MVTec AD — A Comprehensive Real-World Dataset for Unsupervised Anomaly Detection. CVPR, 2019

---

> > ### Author Response · Authors · 2024-11-22
> >
> > ### Image AUROC
> > | **Class Name** | **RealNet** | **RealNet+MADCluster** | **RD4AD** | **RD4AD+MADCluster** | **PyramidFlow** | **PyramidFlow+MADCluster** |
> > |:----------------:|:-------------:|:------------------------:|:-----------:|:-----------------------:|:-----------------:|:----------------------------:|
> > | **Bottle**     | 0.918       | 0.961                 | 0.991     | 0.998                | 0.778           | 0.993                     |
> > | **Cable**      | 0.631       | 0.669                 | 0.945     | 0.946                | 0.638           | 0.695                     |
> > | **Capsule**    | 0.694       | 0.698                 | 0.868     | 0.870                | 0.870           | 0.916                     |
> > | **Carpet**     | 0.969       | 0.977                 | 0.996     | 0.997                | 0.938           | 0.964                     |
> > | **Grid**       | 0.872       | 0.875                 | 0.921     | 0.945                | 0.794           | 0.824                     |
> > | **Hazelnut**   | 0.972       | 0.994                 | 1.000     | 1.000                | 0.930           | 0.935                     |
> > | **Leather**    | 0.806       | 0.830                 | 1.000     | 1.000                | 0.993           | 0.999                     |
> > | **Metal Nut**  | 0.670       | 0.688                 | 0.995     | 0.996                | 0.735           | 0.742                     |
> > | **Pill**       | 0.823       | 0.844                 | 0.936     | 0.956                | 0.810           | 0.834                     |
> > | **Screw**      | 0.552       | 0.572                 | 0.829     | 0.848                | 0.595           | 0.752                     |
> > | **Tile**       | 0.972       | 0.981                 | 0.993     | 0.994                | 0.994           | 0.995                     |
> > | **Toothbrush** | 0.553       | 0.644                 | 0.997     | 1.000                | 0.944           | 0.947                     |
> > | **Transistor** | 0.659       | 0.660                 | 0.967     | 0.970                | 0.908           | 0.936                     |
> > | **Wood**       | 0.959       | 0.966                 | 0.990     | 0.993                | 0.991           | 0.996                     |
> > | **Zipper**     | 0.882       | 0.901                 | 0.871     | 0.889                | 0.938           | 0.938                     |
> >
> > ---
> >
> > ### Pixel AUROC
> > | **Class Name** | **RealNet** | **RealNet+MADCluster** | **RD4AD** | **RD4AD+MADCluster** | **PyramidFlow** | **PyramidFlow+MADCluster** |
> > |:----------------:|:-------------:|:------------------------:|:-----------:|:-----------------------:|:-----------------:|:----------------------------:|
> > | **Bottle**     | 0.949       | 0.963                 | 0.982     | 0.986                | 0.960           | 0.974                     |
> > | **Cable**      | 0.631       | 0.897                 | 0.977     | 0.977                | 0.895           | 0.912                     |
> > | **Capsule**    | 0.901       | 0.927                 | 0.981     | 0.982                | 0.977           | 0.980                     |
> > | **Carpet**     | 0.970       | 0.984                 | 0.992     | 0.992                | 0.964           | 0.978                     |
> > | **Grid**       | 0.873       | 0.894                 | 0.942     | 0.963                | 0.941           | 0.948                     |
> > | **Hazelnut**   | 0.925       | 0.956                 | 0.991     | 0.991                | 0.964           | 0.973                     |
> > | **Leather**    | 0.968       | 0.971                 | 0.994     | 0.994                | 0.985           | 0.987                     |
> > | **Metal Nut**  | 0.754       | 0.770                 | 0.969     | 0.974                | 0.938           | 0.959                     |
> > | **Pill**       | 0.942       | 0.943                 | 0.967     | 0.968                | 0.943           | 0.956                     |
> > | **Screw**      | 0.929       | 0.946                 | 0.985     | 0.986                | 0.898           | 0.903                     |
> > | **Tile**       | 0.930       | 0.937                 | 0.953     | 0.953                | 0.962           | 0.973                     |
> > | **Toothbrush** | 0.918       | 0.924                 | 0.987     | 0.988                | 0.975           | 0.977                     |
> > | **Transistor** | 0.704       | 0.722                 | 0.890     | 0.890                | 0.965           | 0.972                     |
> > | **Wood**       | 0.930       | 0.932                 | 0.955     | 0.955                | 0.957           | 0.960                     |
> > | **Zipper**     | 0.951       | 0.962                 | 0.968     | 0.970                | 0.968           | 0.968                     |
> >
> >
> > We believe this demonstrates the adaptability of MADCluster and its potential to generalize to other domains like image.

---

> > > ### Comment · Reviewer_Gomd · 2024-11-26
> > >
> > > Thank you to the authors for their response and hard work, which effectively addressed my concerns, particularly regarding the application in the image domain. However, as I am less familiar with research in time-series anomaly detection and seem to be an outlier in the scoring, I am not entirely confident with my assessment.

---

### Official Review · Reviewer_3EUZ · 2024-11-03

**Soundness:** 1
**Presentation:** 3
**Contribution:** 2
**Rating:** 3
**Confidence:** 4

**Summary:**

The authors propose MADCluster, a model-agnostic anomaly detection framework by self-supervised clustering with a one-directed adaptive loss. The authors also address the hypersphere collapse problem for clustering-based anomaly detection methods.

**Strengths:**

1. The authors propose MADCluster, a model-agnostic anomaly detection framework.
2. The authors propose a one-directed adaptive loss for single clustering.
3. The authors address the hypersphere collapse problem in the clustering-based anomaly detection methods.

**Weaknesses:**

1. The claimed novelty in this paper is Model-Agnostic. But the authors did not compare with any Model-Agnostic anomaly detection models, such as [1] and [2]. The authors should include these related works in one section, compare these prior works by experiments, and provide a detailed comparison highlighting the key technical differences and innovations compared to these previous works.

    [1] Towards Lightweight, Model-Agnostic and Diversity-Aware Active Anomaly Detection. ICLR 2023.

    [2] Distribution Agnostic Symbolic Representations for Time Series Dimensionality Reduction and Online Anomaly Detection. TKDE 2023.

2. There are no experiments to show or theoretical proof of why updating central coordinates through network parameters can address the hypersphere collapse problem. More experiments or analyses would help demonstrate how or why updating central coordinates prevents hypersphere collapse, for example, using some quantitative metrics and ablation studies to show this effect.

3. The authors argue that they address the hypersphere collapse problem by preventing the all-zero parameter problem. However, the hypersphere collapse problem is not only the all-zero parameter problem but also related to the mode collapse problem, as shown in [3] and [4]. The claim from this paper is not sound. The authors should include these related works in one section, compare these prior works by experiments, and provide a detailed comparison highlighting the key technical differences and innovations of their method.

    [3] Exploring Simple Siamese Representation Learning. CVPR 2021.

    [4] MHCCL: Masked Hierarchical Cluster-Wise Contrastive Learning for Multivariate Time Series. AAAI 2023.

4. The main weakness of this paper is that the authors use the wrong evaluation metrics. The authors use the point adjustment (PA) for evaluation. Many works [5, 6, 7] have demonstrated that PA can lead to faulty performance evaluations, where PA use true labels from the test datasets to adjust the outputs of models, and it is known that using PA can result in state-of-the-art performance even with random scores or random initialized non-trained models [6, 7], making it impossible to conduct a fair comparison and assess the effectiveness of the models. The showed effectiveness in this paper is flawed. The authors should use alternative evaluation metrics, such as original F1, ROC-AUC, PR-AUC, Aff Recall, Precision and F1 [6, 7], instead of PA. The authors should re-run their experiments with these reasonable metrics and discuss how it impacts their results and conclusions.

    [5] Current Time Series Anomaly Detection Benchmarks are Flawed and are Creating the Illusion of Progress. TKDE 2023.

    [6] Drift doesn't Matter: Dynamic Decomposition with Diffusion Reconstruction for Unstable Multivariate Time Series Anomaly Detection. NeurIPS 2023.

    [7] Local Evaluation of Time Series Anomaly Detection Algorithms. KDD 2022.

**Questions:**

see weaknesses.

---

> ### Author Response · Authors · 2024-11-23
>
> We deeply value reviewer 3EUZ’s constructive comments, which have guided us in identifying areas for improvement and further refinement of our research.
>
> **W1**
> > **The claimed novelty in this paper is Model-Agnostic. But the authors did not compare with any Model-Agnostic anomaly detection models, such as [1] and [2]. The authors should include these related works in one section, compare these prior works by experiments, and provide a detailed comparison highlighting the key technical differences and innovations compared to these previous works.**
>
> * We greatly appreciate your insightful suggestion regarding the comparison with model-agnostic anomaly detection. Your feedback highlights an important aspect that we acknowledge could strengthen our work.
>
>
> * While our primary objective was to address the hypersphere collapse issue explicitly mentioned in [1], we recognize that the evaluation of our model focused more on comparisons with general model-agnostic approaches, rather than directly addressing methods tailored to this specific challenge. This observation provides a valuable perspective for broadening the relevance and impact of our study.
>
>     [1] Deep One-Class Classification. PMLR, 2018.
>
>
> * In response, we plan to include a dedicated section in the revised manuscript discussing these prior works, conduct experimental comparisons with them, and provide a comprehensive analysis of the key technical differences and innovations of our approach in relation to these methods. We believe this will substantially enhance the clarity and rigor of our contributions.

---

> > ### Author Response · Authors · 2024-11-23
> >
> > **W2**
> > > **There are no experiments to show or theoretical proof of why updating central coordinates through network parameters can address the hypersphere collapse problem. More experiments or analyses would help demonstrate how or why updating central coordinates prevents hypersphere collapse, for example, using some quantitative metrics and ablation studies to show this effect.**
> >
> > * We believe that **Figure 4** effectively illustrates why updating the central coordinates through network parameters helps address the hypersphere collapse problem. Furthermore, the process of demonstrating the effectiveness of updating central coordinates, along with the validation of the proposed one-directed adaptive loss, is thoroughly detailed in **Appendix A**.
> >
> > * The proposed one-directed adaptive loss is theoretically supported to perform better in classifying anomalies.
> >     - **Criteria for distinguishing normal data and anomaly data (Cosine similarity based)**
> >         - We calculated the cosine similarity $q_t$ between the embedded $h^f_t$ in the feature space and the cluster center coordinate $\hat{c}$ using **equation (3)**. Then we applied the transformation $(q_t+1)/2$, adjusting the range of $q_t$ to lie between 0 and 1.
> >         - This approach means that as $q_t$ approaches 1, the data representation on feature space aligns more closely with the normal cluster center. Consequently, when $q_t$ is near 1, the data can be classified as normal, whereas values closer to 0 indicate an anomalous classification. Specifically, for the two discrete labels 0 and 1, values of $q_t$ above a defined threshold $\nu$ are labeled as normal, while values below this threshold indicate anomaly.
> >     - **Adapting Thresholds During Normal Data Training**
> >         - Meanwhile, the objective of the distance loss **equation (2)** is to enclose a significant amount of normal data while minimizing the hypersphere radius $R$. As the learning process of the distance loss progresses, the criteria for classifying data as normal become more stringent, meaning the threshold $\nu$ in **equation (4)** should be gradually increased. Since our method learns only from normal data, both the data labeled as 1 and the increasing of threshold $\nu$ in the one-directed adaptive loss **equation (5)** have to adapt accordingly as learning continues.
> >     - **Summary and application in anomaly detection**
> >         - In summary, from a theoretical perspective, normal data distribution becomes more concentrated around the cluster center coordinates $\hat{c}$ under one-directed adaptive loss **equation (5)**, compared to models without this loss function.
> >         - We assume that the value of $q_t$ corresponding to the anomaly data will be significantly less than 1 when new anomaly data comes in. This result in the learning approach effectively distancing anomalous data outside the hypersphere, thus aiding in anomaly detection. We think **Figure 4** sufficiently reflects this trend, even though it's qualitative.
> >
> >
> > - Ultimately, the proposed one-directed adaptive loss functions to ensure that normal data becomes more densely concentrated around the cluster center coordinate $\hat{c}$.
> >
> >
> > - This outcome is reflected in the experimental results presented in **Table 4** of **Appendix C.2.** In **Table 4**, the results for DeepSVDD represent the use of $\mathcal{L}\*{\text{distance}}$ with fixed center coordinates only, while the results for Clustering reflect the use of $\mathcal{L}\*{\text{cluster}}$, which is a one-directed adaptive loss, without employing $\mathcal{L}\*{\text{distance}}$. Finally, MADCLUSTER incorporates both $\mathcal{L}\*{\text{distance}}$ and $\mathcal{L}\_{\text{cluster}}$.

---

> > > ### Author Response · Authors · 2024-11-23
> > >
> > > **W3**
> > > > **The authors argue that they address the hypersphere collapse problem by preventing the all-zero parameter problem. However, the hypersphere collapse problem is not only the all-zero parameter problem but also related to the mode collapse problem, as shown in [3] and [4]. The claim from this paper is not sound. The authors should include these related works in one section, compare these prior works by experiments, and provide a detailed comparison highlighting the key technical differences and innovations of their method.**
> > >
> > > * We sincerely thank the reviewer for raising the important point regarding mode collapse. While we acknowledge that mode collapse is a relevant concern, as previously mentioned, our proposed method explicitly addresses the hypersphere collapse problem. However, we would like to clarify how our approach also prevents mode collapse
> > >
> > >
> > >  1. Without the one-directed adaptive loss, the movement of the central coordinates may allow for good representation power. However, this could introduce risks not only of hypersphere collapse but also of mode collapse.
> > >
> > >
> > > 2. When the one-directed adaptive loss is incorporated, learning is guided such that the threshold $\nu$ increases, resulting in the central coordinates being updated in a direction that reduces the proportion of data classified as normal. This gradual reduction effectively refines the clustering of normal data.
> > >
> > >
> > > 3. If mode collapse were to occur, it would hinder performance improvements. However, as shown in **Table 1**, the application of MADCluster demonstrates consistent performance improvements, indicating that the model avoids mode collapse. Instead, the model learns a rich representation of complex normal data, effectively distinguishing between normal and anomalous samples. We believe this qualitative improvement is well-illustrated in **Figure 3** and **Figure 4**.
> > >
> > >
> > > **W4**
> > > > **The main weakness of this paper is that the authors use the wrong evaluation metrics. The authors use the point adjustment (PA) for evaluation. Many works [5, 6, 7] have demonstrated that PA can lead to faulty performance evaluations, where PA use true labels from the test datasets to adjust the outputs of models, and it is known that using PA can result in state-of-the-art performance even with random scores or random initialized non-trained models [6, 7], making it impossible to conduct a fair comparison and assess the effectiveness of the models. The showed effectiveness in this paper is flawed. The authors should use alternative evaluation metrics, such as original F1, ROC-AUC, PR-AUC, Aff Recall, Precision and F1 [6, 7], instead of PA. The authors should re-run their experiments with these reasonable metrics and discuss how it impacts their results and conclusions.**
> > >
> > > * We sincerely appreciate your detailed feedback regarding the evaluation metrics used in our study, particularly the concerns surrounding point adjustment (PA). We primarily used PA in our experiments because of its widespread use in prior works [1, 2, 3, 4, 5] as a standard performance metric for model comparisons.
> > >
> > >     [1] Unsupervised Anomaly Detection via Variational Auto-Encoder for Seasonal KPIs in Web Applications. WWW, 2018.
> > >
> > >     [2] Robust Anomaly Detection for Multivariate Time Series through Stochastic Recurrent Neural Network. KDD, 2019.
> > >
> > >     [3] Timeseries Anomaly Detection using Temporal Hierarchical One-Class Network. NeurIPS, 2020.
> > >
> > >     [4] Anomaly Transformer: Time Series Anomaly Detection with Association Discrepancy. ICLR, 2022.
> > >
> > >     [5] DCdetector: Dual Attention Contrastive Representation Learning for Time Series Anomaly Detection. KDD, 2023.
> > >
> > > - Your insights are highly appreciated and will contribute significantly to enhancing the quality and validity of our research.

---

> ### Comment · Reviewer_3EUZ · 2024-11-25
> **Official Comment by Reviewer 3EUZ**
>
> The author’s response addresses most of my questions.
>
> However, I remain unconvinced by the answer to w1 and w4.
>
> w1. The author should include analysis and especially include experiments with other Model-Agnostic anomaly detection models, not limited to [1] and [2] but also other methods that address hypersphere collapse issues.
>
> w4. Existing works having problems do not mean that you can continue to make mistakes.
>
> Many issues regarding the Point Adjust are discussed after these works. See https://github.com/thuml/Anomaly-Transformer/issues/14 and https://github.com/thuml/Anomaly-Transformer/issues/65 for example. Many works, not limited to [5, 6, 7], have demonstrated that PA can lead to faulty performance evaluations. Using PA can result in state-of-the-art performance even with random scores or random initialized non-trained models, making it impossible to conduct a fair comparison and assess the effectiveness of the models. I insist that the showed effectiveness in this paper is flawed.  I insist that the authors should use alternative reasonable evaluation metrics, such as original F1, ROC-AUC, PR-AUC, Aff Recall, Precision and F1 [6, 7], instead of PA.

---

### Official Review · Reviewer_rPiY · 2024-11-08

**Soundness:** 2
**Presentation:** 3
**Contribution:** 2
**Rating:** 5
**Confidence:** 3

**Summary:**

This paper targets on the model-agnostic anomaly detection
for time series and try to addresses the ‘hypersphere collapse’ problem in existing deep learning-based self-supervised clustering anomaly detection methods. The authors introduce learnable cluster center and one-directed adaptive loss to improve the Cluster Distance Mapping module and Sequence-wise Cluster module. The experiments on four time series benchmark datasets showcase that their method can further improve the performance of based model.

**Strengths:**

1. The paper is well-written and easy to follow.
2. The motivation of learnable cluster center and one-directed adaptive loss is clear and well proved through the experiments.

**Weaknesses:**

1. The paper does not present sufficient discussion of the reconstruction-based methods, which should be a very important type of methods in time series anomaly detection. More speficially, it seems that the authors can conbine their method with the reconstruction-based methods such as Anomaly Transformer and DCdetector. I wonder how to implement such conbination. Do you only use the network backbones from the Anomaly Transformer or DCdetector, or delivers a reconstruction loss (or even contrastive loss) together with your loss function?
2. There is no ablation study in the paper to demonstrate the independent function of learnable cluster center and one-directed adaptive loss.
3. Some descriptions are inconsistent. For example, from line 221-225, only the neural network weights W and the cluster center parameters c are learnable. However, from the experiments, the radius R is also learnable. Additionally, it is quite strange to optimize c seperately with equation (6) but use adam to optimze other parameters.
4. The discussion for one-directed adaptive loss function is not so clear and sufficient in the main paper. Even though the detailed explanation is provided in the appendix, the authors are suggested to provide a short but more clear and intuitive explanation about how equation (4) is developed in the main paper.
5. More comparison with the pretrained methods for TS anamoly detection should be discussed, such as TimesNet[1] and FPT[2].

[1] Wu, Haixu, et al. "TimesNet: Temporal 2D-Variation Modeling for General Time Series Analysis." The Eleventh International Conference on Learning Representations.
[2] Zhou, Tian, et al. "One fits all: Power general time series analysis by pretrained lm." Advances in neural information processing systems 36 (2023): 43322-43355.

**Questions:**

1. How to implement your method with reconstruction-based anamaly detection methods?
2. Any ablation study for learnable cluster center and one-directed adaptive loss?
3. It is possbile to implement your method with some pre-trained TS foundation models?

---

> ### Author Response · Authors · 2024-11-21
>
> We sincerely thank the reviewer rPiY for valuable and constructive feedback.
>
> **Q1 & W1**
> > **How to implement your method with reconstruction-based anomaly detection methods?**
>
> * In our method, we incorporate not only the network backbones from reconstruction-based methods such as Anomaly Transformer and DCdetector but also leverage their reconstruction loss alongside our proposed loss function. Specifically, we integrate the MADCluster module into the official implementations of these methods. The MADCluster loss is computed and added as an additional term to the final loss function.
>
> * The integration process is tailored to the specific network backbone, with a critical consideration being the selection of appropriate data inputs for the MADCluster module. In our experiments, we used the embeddings generated by the DataEmbedding class in both Anomaly Transformer and DCdetector as inputs to the MADCluster module.
>
> **Q2 & W2**
> > **Any ablation study for learnable cluster center and one-directed adaptive loss?**
>
> * We would like to kindly point out that **Table 4** in **Appendix C.2** presents the results of an ablation study.
>
>
> * We did not conduct experiments on the learnable cluster center without applying the one-directed adaptive loss because, as indicated in [1], including $c$ as an optimization variable leads to a trivial solution known as 'hypersphere collapse.' If $c$ is learned along with the network weights, the network can minimize the objective function by mapping all inputs to $c$. This results in a hypersphere with a radius of 0, preventing the network from learning meaningful representations.
>
> * Therefore, we applied the one-directed adaptive loss to enable the learnable cluster center. In **Table 4**, the results for DeepSVDD represent the use of $\mathcal{L}\_{\text{distance}}$ with fixed center coordinates only, while the results for Clustering reflect the use of $\mathcal{L}\_{\text{cluster}}$, which is a one-directed adaptive loss, without employing $\mathcal{L}\_{\text{distance}}$. Finally, MADCLUSTER incorporates both $\mathcal{L}\_{\text{distance}}$ and $\mathcal{L}\_{\text{cluster}}$.
>
>
>    [1] Deep One-Class Classification. PMLR, 2018.
>
> **Q3 & W5**
> > **It is possible to implement your method with some pre-trained TS foundation models?**
>
> * The applicability of our method does not depend on specific pre-trained foundation models. Rather, if the embeddings generated within a model can be utilized, the MADCluster can be applied to calculate the loss or anomaly score. However, when using pre-trained foundation models, it is necessary to train the MADCluster module with normal data only to ensure proper integration and functionality.
>
>
> **W3**
> > **Regarding the comment: “Some descriptions are inconsistent. For example, from line 221-225, only the neural network weights $\mathcal{W}$ and the cluster center parameters $c$ are learnable. However, from the experiments, the radius $R$ is also learnable.”**
>
> * We have clarified this in our revised submission to avoid potential misunderstandings. Specifically, $R$ is not a learnable parameter; rather, it is determined based on the neural network outputs and the given hyperparameter $\rho$. $R$ is computed as a specific quantile of the neural network outputs and the loss values. This clarification has been added to the manuscript to ensure consistency in the description.
>
> > **Regarding the comment: “However, from the experiments, the radius $R$ is also learnable.”**
>
> * As shown in the results of the second column in **Figure 5** of the paper, the change in the radius $R$ is not due to it being updated as a parameter. Instead, as previously mentioned, $R$ is visualized as being reduced based on a specific quantile value of the computed loss.
>
> > **Regarding the comment: “Additionally, it is quite strange to optimize $c$ separately with **equation (6)** but use Adam to optimize other parameters.”**
>
> * Thank you for pointing this out. We acknowledge that the description of the update method for the center coordinates $c$ was overly generalized by representing it as an SGD approach. To address this, we have eliminated **equation (6)** in the previous manuscript to more accurately reflect the update process. Additionally, we have removed the reference to Adam in **Algorithm 1** to ensure consistency and clarity in the description of the optimization methods.

---

### Author Response · Authors · 2024-11-28

Dear Reviewers,

First and foremost, we would like to sincerely thank all the reviewers for their valuable suggestions and constructive feedback on our paper. I truly appreciate the time and effort you have taken to provide these insightful comments.

We have made every effort to address all the weaknesses and questions raised, aiming to clarify any misunderstandings and improve the overall quality of my work. However, we must acknowledge that it might be challenging to provide experimental results within the rebuttal period due to time constraints.

Nonetheless, we are committed to conducting these additional experiments and incorporating all the feedback into our research to ensure a more thorough and comprehensive study.

Once again, thank you very much for your thoughtful comments and for helping us.

---

### Meta-Review · Area_Chair_KY2m · 2024-12-18

**Metareview:**

**Overview**. The paper introduces a method that utilizes a self-supervised clustering method with learnable prototypes for time series anomaly detection. The method is evaluated on four TSAD datasets and shows effective detection performance.

**Strengths**.
- The paper is well written and easy to follow [rPiY, Gomd]
- The effectiveness of the proposed clustering-based method is demonstrated by the experimental results on the datasets used [rPiY, wgG7, VmN2]
- The proposed method helps address the notorious hypersphere collapse issue [3EUZ, VmN2]

**Weaknesses**.
- It is difficult to distinguish the method from closely related reconstruction-based methods or enhanced SVDD methods [rPiY, VmN2]
- Ablation study is not convincing [rPiY, 3EUZ, wgG7, VmN2]
- Comparison to related and/or recent state-of-the-art TSAD methods, such as pre-trained TS foundation models, model-agnostic methods or self-supervised methods, is missing [rPiY, 3EUZ, wgG7]
- The justification on addressing hypersphere collapse is weak [3EUZ, wgG7]
- More evaluation metrics should be used [3EUZ, wgG7]
- Lack of computational runtime analysis [Gomd]
- The performance improvement is marginal compared to the best competing methods [VmN2]

**Additional Comments On Reviewer Discussion:**

The work has five reviewers, four of which engaged with the authors during discussion. The rebuttal helps address some of the concerns, such as the generalization of the method from time series data to image data, concerns on the proposed loss function and justification on addressing the hypersphere collapse, ablation study on the loss function, etc. One reviewer increased the rating from weak reject to weak accept, while the other four reviewers did not change their ratings, one reject and three weak rejects. The concerns raised again in the discussion include the lack of empirical comparison to related model-agnostic methods, evaluation metrics issue, marginal improvement over the competing methods, etc. Overall, while the method shows some interesting generalization results, the paper can be significantly enhanced with more comprehensive evaluation on diverse data types and evaluation metrics to provide a strong justification on the model-agnostic characteristic. The justification on addressing the hypersphere collapse should be strengthened too, e.g., with more explicit discussion and comparison to existing solutions. Thus, the AC finds that the paper is not ready for publication at ICLR in its current form.

---

### Decision · Program_Chairs · 2025-01-22

Reject